# Molecular insights into biogenesis of glycosylphosphatidylinositol anchor proteins

Yidan Xu[1,5], Guowen Jia[2,5], Tingting Li ⬤ [1,5], Zixuan Zhou[3,5], Yitian Luo ⬤ [1,4], Yulin Chao[3], Juan Bao[1], Zhaoming Su ⬤ [2✉], Qianhui Qu ⬤ [3✉] & Dianfan Li ⬤ [1✉]

Eukaryotic cells are coated with an abundance of glycosylphosphatidylinositol anchor proteins (GPI-APs) that play crucial roles in fertilization, neurogenesis, and immunity. The removal of a hydrophobic signal peptide and covalent attachment of GPI at the new carboxyl terminus are catalyzed by an endoplasmic reticulum membrane GPI transamidase complex (GPI-T) conserved among all eukaryotes. Here, we report the cryo-electron microscopy (cryo-EM) structure of the human GPI-T at a global 2.53-Å resolution, revealing an equimolar heteropentameric assembly. Structure-based mutagenesis suggests a legumain-like mechanism for the recognition and cleavage of proprotein substrates, and an endogenous GPI in the structure defines a composite cavity for the lipid substrate. This elongated active site, stemming from the membrane and spanning an additional ~22-Å space toward the catalytic dyad, is structurally suited for both substrates which feature an amphipathic pattern that matches this geometry. Our work presents an important step towards the mechanistic understanding of GPI-AP biosynthesis.

[1] State Key Laboratory of Molecular Biology, CAS Center for Excellence in Molecular Cell Science, Shanghai Institute of Biochemistry and Cell Biology, University of CAS, Chinese Academy of Sciences (CAS), 320 Yueyang Road, 200030 Shanghai, China. [2] State Key Laboratory of Biotherapy and Cancer Center, Department of Geriatrics and National Clinical Research Center for Geriatrics, West China Hospital, Sichuan University, 610044 Chengdu, China. [3] Shanghai Stomatological Hospital, School of Stomatology, Shanghai Key Laboratory of Medical Epigenetics, International Co-laboratory of Medical Epigenetics and Metabolism (Ministry of Science and Technology), Institutes of Biomedical Sciences, Department of Systems Biology for Medicine, Fudan University, 200032 Shanghai, China. [4] School of Life Science and Technology, ShanghaiTech University, 393 Middle Huaxia Road, 201210 Shanghai, China. [5] These authors contributed equally: Yidan Xu, Guowen Jia, Tingting Li, Zixuan Zhou. ✉email: zsu@scu.edu.cn; qqh@fudan.edu.cn; dianfan.li@sibcb.ac.cn

The GPI anchoring represents a ubiquitous, metabolically expensive posttranslational modification of eukaryotic cell surface proteins[1–5]. Structurally elucidated in the 1980s[6], GPI lipids are bioactive[7] and chemically diverse with a minimal backbone consisting of a phosphatidylinositol group linked to a polysaccharide core α-Man3-(1 → 2)-α-Man2-(1 → 6)-α-Man1-(1 → 4)-α-GlcN (Man1/2/3, the three mannoses; GlcN, glucosamine; numbers in parentheses indicate carbon numbering) (Fig. 1a, Supplementary Fig. 1a). The glycan core in the mature GPI carries additional functional groups, some of which are species- and tissue-specific. The maturation process involves multiple enzymes including mannosyl transferases and phosphorylethanolamine (EtNP) transferases (Supplementary Fig. 1b)[2–4]. The EtNP on C2 of Man1 is a prerequisite for the enzymatic addition of Man3[8,9], after which step two EtNPs are sequentially transferred to C6 of Man3 (as the linker for GPI anchoring) and Man2 (for efficient endoplasmic reticulum (ER)-Golgi transport of some GPI-APs)[2]. The mannosylation of Man3 at C2 is essential for GPI anchoring in some species like yeasts[10]. In some Trypanosoma species, the C3 of Man2, and C3/C4 of Man1 can have further saccharide decorations and the C6 of GlcN is modified with an aminoethylphosphonate (summarized in ref. [11]) (Supplementary Fig. 1a). Regarding the phosphatidylinositol part, an acyl chain is usually present at C2 of inositol before GPI anchoring (Supplementary Fig. 1a) but is in most cases (except in erythrocytes) removed immediately after GPI attachment[3]. Finally, the fatty chain of the phosphatidyl group also undergoes remodeling both before (Supplementary Fig. 1b) and after the GPI-attachment step, causing acyl diversity such as diacylglycerol, alkylacylglycerol, or ceramides with varying length and unsaturation[2–4,11].

These complicated anchorages place GPI-APs in lipid rafts and confer their unique regulatory properties in developmental and physiological processes[2–5]. Notable GPI-APs include LY6K/TEX101 as key factors for fertilization and biomarkers for infertility[12], glypicans/Gas1/RECK that modulate Hedgehog/Wnt/Notch signaling[13,14], CD55/CD59 that inhibit complement cascade in innate immunity[15], alkaline phosphatase as a leading biomarker for hepatic diseases and cancers[16], and folate receptor 1 that mediates folate uptake and as an important cancer biomarker[17]. Disruption of the GPI-AP biosynthesis leads to embryonic lethality in animals[18], while its biogenesis pathway in *Trypanosoma brucei* is a validated drug target for the fatal sleeping sickness[19].

The committed step in GPI-AP biogenesis is catalyzed by the GPI transamidase (GPI-T), which cleaves the C-terminal signal peptide (CSP) of precursor protein and covalently links the EtNP moiety of the mature GPI (typically in the form of the GPI core with three EtNPs and three acyl chains, Fig. 1a) to the newly exposed carboxyl terminus of the so-called ω-residue (Fig. 1a)[1,2,20]. The ω-site is not a specific residue but rather has a small site chain such as glycine, alanine, serine, cysteine, aspartate, and asparagine. The CSP also lacks sequence consensus but rather contains a pattern with a C-terminal hydrophobic tail (15–20 residues) connecting to the ω-site via a hydrophilic spacer (8–12 residues) with a small side chain at ω+2 position (Fig. 1a). GPI-T consists of at least five subunits (Supplementary Fig. 1c), namely PIGK/Gpi8p, PIGT/Gpi16p, PIGU/Gab1p, PIGS/Gpi17p, GPAA1/Gaa1p in human/yeast, respectively (Supplementary Fig. 2)[2]. For ease of description, we use the human nomenclature hereafter. PIGK and GPAA1 have been proposed to execute the peptide cleavage[21,22] and GPI addition reactions[23]. PIGU and GPAA1 have been suggested to bind GPI[20,24]. PIGT disulfide-links with PIGK in some species[25] and may play a structural role. The function of PIGS is less clear although it is essential for GPI-T activity[26]. All subunits are predicted to contain at least one

transmembrane helix (TMH) except for PIGK in some species like *T. brucei*[27]. Aberrant activity of GPI-T has been recently implicated in multiple pathologies[28–37] such as neurodevelopmental disorder with hypotonia and cerebellar atrophy, with or without seizures (NEDHCAS) and cancers. No experimental structural information exists to explain and clarify various and sometimes contradictory models for GPI-T's assembly[26,38,39], membrane topology[24,40], and subunit function[20–24].

Here, we determine a global 2.53Å-resolution cryo-EM structure of the human glycosylphosphatidylinositol transamidase (GPI-T), revealing an equimolar heteropentameric architecture. Structural comparison and mutagenesis suggest a legumain-like mechanism for peptide cleavage. The catalytic site is positioned ~22 Å away from the membrane interface, a characteristic geometry that may confer specificity for both the proprotein and the GPI substrates. Densities of a GPI molecule in a shared cavity and rational mutagenesis define an elongated active site that stretches from the membrane to the catalytic dyad. Our work represents an important step towards the mechanistic understanding of the GPI-AP biosynthesis and the pathophysiology associated with GPI-T mutations.

## Results

**Structure determination of the pentameric GPI-T complex**. To gain insights into the GPI anchoring process, we set to determine its 3D structure by starting with its recombinant expression. The human GPI-T subunits were co-expressed in HEK293 cells with a thermostable green fluorescence protein (TGP)[41] tag at the C-termini to facilitate the purification process. The membrane protein complex was solubilized in the detergent lauryl maltose neopentyl glycol (LMNG) and then exchanged into digitonin on a Strep-affinity column via PIGU. A second affinity chromatography via the nonahistidine tag on PIGT was performed to minimize the purification of free subunits. The complex was further fractioned by gel permeation chromatography. The peak fractions contained all five subunits along with minor high-molecular-weight contaminants based on the in-gel fluorescence and Coomassie staining results (Fig. 1b).

Using single-particle cryo-EM, we determined the structure of the transamidase complex at 2.53 Å nominal resolution (Supplementary Fig. 3a–c, Supplementary Table 1). The high-quality map (Supplementary Fig. 3d) sufficed ab initio model building, with a total of 2,393 residues (94.4% completion), 3 N-glycosylation sites, 4 disulfide bonds, and 22 lipid/detergent molecules.

Consistent with the in-gel fluorescence results, the complex structure contained five subunits with 1:1 stoichiometry. The overall GPI-T architecture assumes the shape of a canon (the luminal domain, mainly from PIGT/GPAA1/PIGS/PIGK) on a carriage (the transmembrane domain, TMD, mainly from GPAA1/PIGU) with an approximate dimension of 148 Å by 141 Å by 83 Å (Fig. 1c). The TMD consisting of 24 TMHs is divided to contain a small entity with eight TMHs from GPAA1 and a large entity with a TMH core (TMH1-12 of PIGU) surrounded by four satellite TMHs, two from PIGS and one each from PIGT and PIGK. Apart from direct contacts, the inter-subunit TMH interactions are further strengthened by lipids including the added cholesteryl hemisuccinate and co-purified phospholipids in both membrane leaflets (Fig. 1c).

**A legumain-like active site for the proprotein substrate**. In the transamidation process, GPI-T cleaves the proprotein at the ω-residue for GPI attachment. This peptide scission is widely believed to be carried out by the PIGK subunit, a cysteine protease of the C13 family whose members include legumains and caspases[21,22]. Indeed, the protease domain of PIGK shares modest

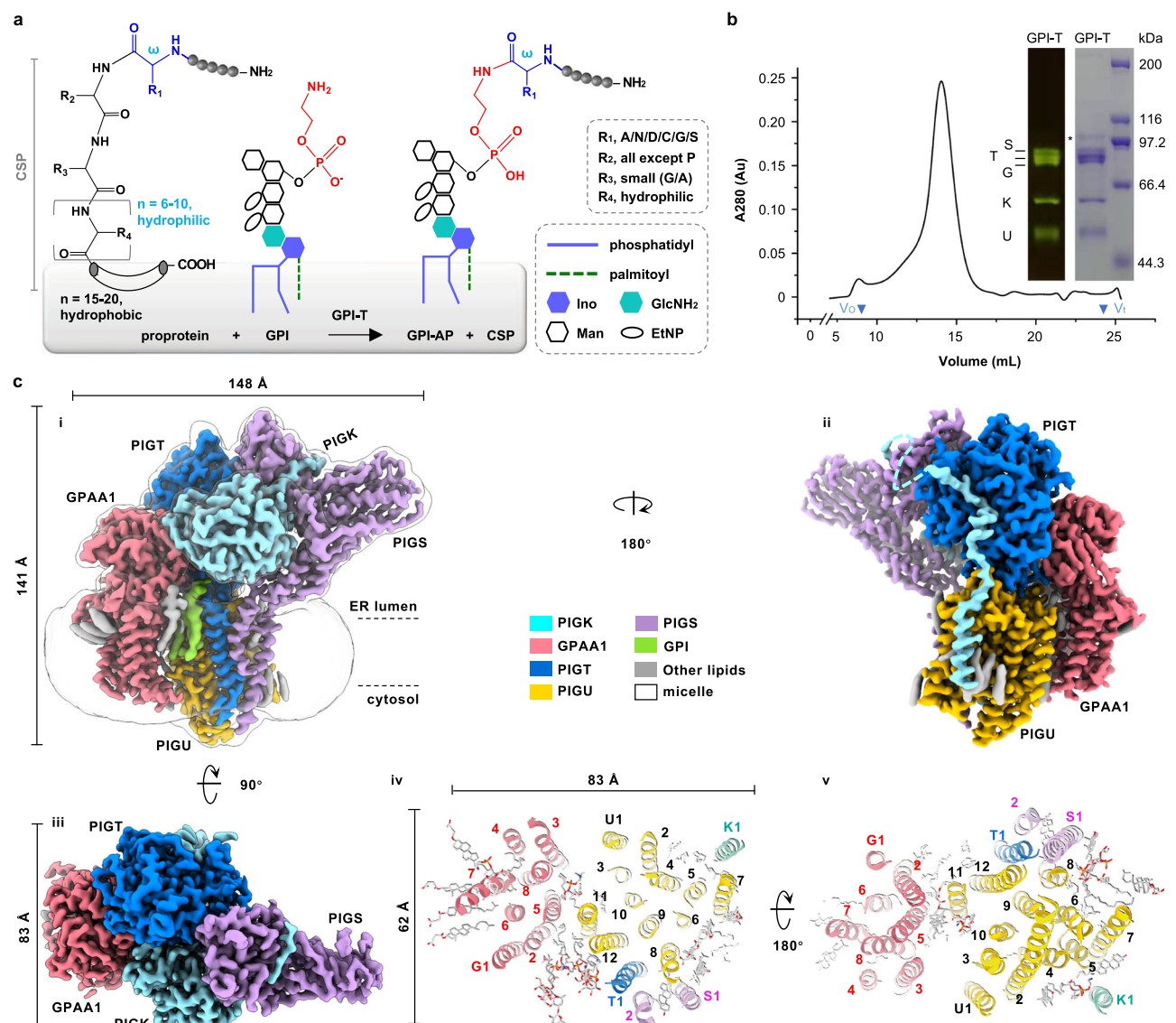

**Fig. 1 Cryo-EM map of the human GPI-T. a** GPI-T replaces the C-terminal signal peptide (CSP) of proproteins with GPI at the ω residue (blue) by a transamination reaction. Various parts are denoted in the dashed box. EtNP, ethanolamine phosphate; Man, mannose; Ino, inositol; GlcNH2, glucosamine. The preferences for the ω, ω+1, ω+2, and ω+3 sites are indicated in the dashed box with amino acid abbreviated with single letters. The gray shading indicates the endoplasmic reticulum (ER) membrane. **b** GPI-T shows a near-Gaussian peak on gel filtration and all five subunits are present on an SDS-PAGE (inset) visualized by in-gel fluorescence (left) and Coomassie staining (right). $V_o$ and $V_t$ (triangle) indicate void and total volume, respectively. Background absorbance signals before $V_o$ are not shown fully. G/T/S/K/U refers to the subunits GPAA1/PIGT/PIGS/PIGK/PIGU. The position of each subunit was separately determined by comparing the complex with singly expressed subunits. An asterisk indicates a minor contaminant. Molecular weights of the protein markers are indicated on the right. Shown is a representative result of three independent experiments. Uncropped images are provided in Source Data. **c** Cryo-EM map (**i-iii**) and normal view of the transmembrane domain from ER lumen (**iv**) or cytosol (**v**). Numbers in **iv** and **v** indicate TMHs and G/T/U/S/K refer to GPAA1 and PIGT/U/S/K, respectively. Lipids and detergents are shown as sticks (gray). Subunits and associated cryo-EM densities are color-coded as indicated.

sequence homology (28.5% identity, 46.4% similarity) (Supplementary Fig. 4a) and high structural homology with legumains (Cα-RMSD of 2.3 Å) (Supplementary Fig. 4b–e)[42], featuring a central 6-stranded β-sheet with 3 α-helices on either side (Supplementary Fig. 4b, d, e). Furthermore, elements essential for legumains' protease activity, including the catalytic dyad, the trivalent oxyanion hole, and the residues in the substrate-binding pockets S1/S1′/S2′, are mostly conserved and superimposable (Fig. 2a, Supplementary Fig. 4d, e) between the two.

To ascertain the functional importance of the structural similarity at the active site, we used a cell-based GPI-AP reporter assay[24] to probe the apparent activity of the mutants. In this

assay, the surface display of the reporter CD59 was monitored by flow cytometry in GPI-T knockout HEK293 cells upon ectopic expression of mutants. As a control, mutations of the previously identified catalytic dyad[21] (H164A or C206S) abolished GPI-T activity (Fig. 2b).

The mutagenesis identified critical residues at the active site and suggested their possible contributions. Substituting R60 at the S1 site with glutamate almost diminished GPI-T activity (9.8% relative to the wild-type (WT) and same hereafter), and R60A, R60L, or even the conservative R60K mutation reduced the activity by 20–60% (Fig. 2b, Supplementary Fig. 5a), suggesting that both the shape and the positive charge of R60 were

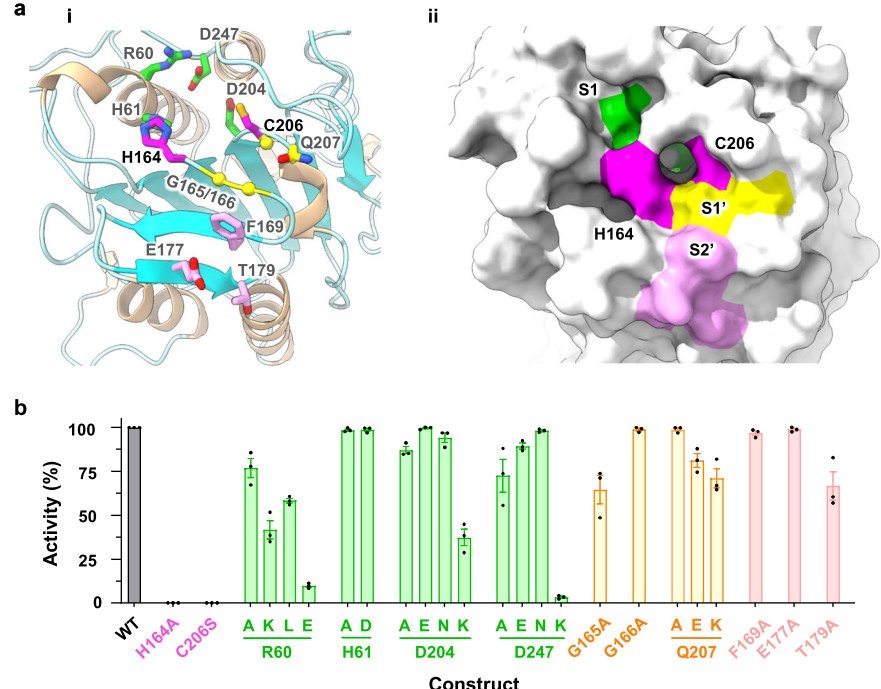

**Fig. 2 Structural and functional resemblance of the PIGK active site to that of legumains. a** Cartoon (wheat, α-helix; cyan, β-strand, **i**) and surface (**ii**) representation of the PIGK protease domain with the catalytic dyad (magenta), S1 (green), S1′ (yellow), and S2′ (pink) residues shown as sticks (**i**) or highlighted in colors (**ii**). S1/S1′/S2′ sites are superposed from legumain structures[44]. **b** Functional assay of the active site mutants. Apparent activity (% of wild-type, WT) was measured by immune staining of a reporter GPI-AP (CD59) on the surface of PIGK-KO cells transfected with the indicated mutants of the catalytic dyad and S1/S1′/S2′ residues (color-coded to match those in **a**). Cells expressing PIGK fused with a thermostable green fluorescence protein[41] were gated, and the surface staining of CD59 was further analyzed by flow cytometry. Data represent mean ± s.e.m. from three independent experiments. Source data are provided.

important for the integrity of S1. The negative charge on D204 or D247 was not essential (D204N, 94.2%; D247N, 98.2%) but a charge reversal was detrimental (D204K, 37.3%; D247K, 3.2%). In contrast, H61 tolerated mutation to alanine or even aspartate (H61A, 98.5%; H61D, 98.7%) (Fig. 2b), suggesting a less important role. The mutagenesis results also indicate the functional importance of three residues in the corresponding S1′ (G165A, 64.5%; Q207E, 81.2%, Q207K, 71.2%) and S2′ site (T179A, 66.7%) (Fig. 2b)[43,44]. Taken together, PIGK may use a similar mechanism as legumains for peptide cleavage and even substrate recognition.

**A composite GPI-binding site**. The GPI attachment to the exposed ω-residue by GPI-T has been proposed to occur concomitantly with the peptide cleavage step[45], suggesting that the GPI-binding site is proximal to the PIGK catalytic dyad. Carrying three acyl chains (Fig. 1a), this lipid substrate is expected to be rooted in the membrane within ~25 Å of the catalytic dyad based on the approximate length of the glycans between the reactive EtNP3 and the phosphatidyl group (Supplementary Fig. 1a). Intriguingly, densities that fit an almost complete GPI core (palmitoylated phosphatidylinositol, glucosylamine, and EtNP-modified Man1) (Fig. 3a) were observed in a cavity jointly formed by TMH11/12 of PIGU, TMH2 of GPAA1, and the sole TMH of PIGT (Fig. 3b) directly "underneath" the catalytic dyad. Although the precise orientation of the EtNP1 moiety is ambiguous at the current resolution, the density for the characteristic tri-acyl chain, the inositol-glucosylamine-mannose glycan core is well-resolved.

The co-purification of the endogenous GPI ligand suggests its relatively tight binding to the complex. Indeed, in our model, the GPI core adheres to the cavity with a rich network of interactions. Specifically, the acyl chains glue to the transmembrane region

mostly of PIGU, and the head group forms hydrogen bonds with PIGT N461/D521/S523 and PIGU N383/N385 (Fig. 3c).

Interestingly, this composite cavity is additionally filled with a density that fits nicely for a digitonin molecule which also contains a polysaccharide chain like GPI (Supplementary Fig. 6a). Further, several evolutionarily conserved residues (PIGK F55/D174/S175, PIGT Y456/D459/P460/N461/D521/F522/S523/M524/N527) (Supplementary Fig. 6b, c) delineate a surface-exposed patch stretching from the membrane interface to the catalytic dyad (Fig. 3b). Notably, this motif is rich in acidic/aromatic residues that are capable of forming H-bonds/"greasy slides" frequently found in protein-sugar interactions[46]. Making topological sense, the cavity is oriented in a way it would fit a proprotein substrate with the hydrophobic portion of CSP in the membrane while the hydrophilic portion would span the ~22 Å space to reach the active site (Fig. 3b). The composite nature of the GPI-binding site, while has been unrecognized by genetic and biochemical characterizations in previous studies, is in line with the findings from such studies that all GPI-T subunits are essential for GPI anchoring[20,22,26,47].

To further characterize the cavity, we generated 30 mutants targeting 10 residues in the conserved patch and the hydrogen-bonding network involved in GPI-binding. Three residues, two from PIGT and one from GPAA1, responded appreciably to mutagenesis. D521 formed two H-bonds with GPI (Fig. 3c). Consistently, substituting it with an alanine decreased the apparent activity to 28.9%, and introducing intended steric clashes by a leucine reduced the activity to 18.8%. The H-bond by PIGT S523 was seemingly non-essential (S523A, 98.4%), but mutating to bulky residues reduced apparent activity (S523F, 76.5%; S523W, 76.8%) (Fig. 3d) presumably by steric hindrance. Further, a double mutant (D521L/S523F) almost abolished

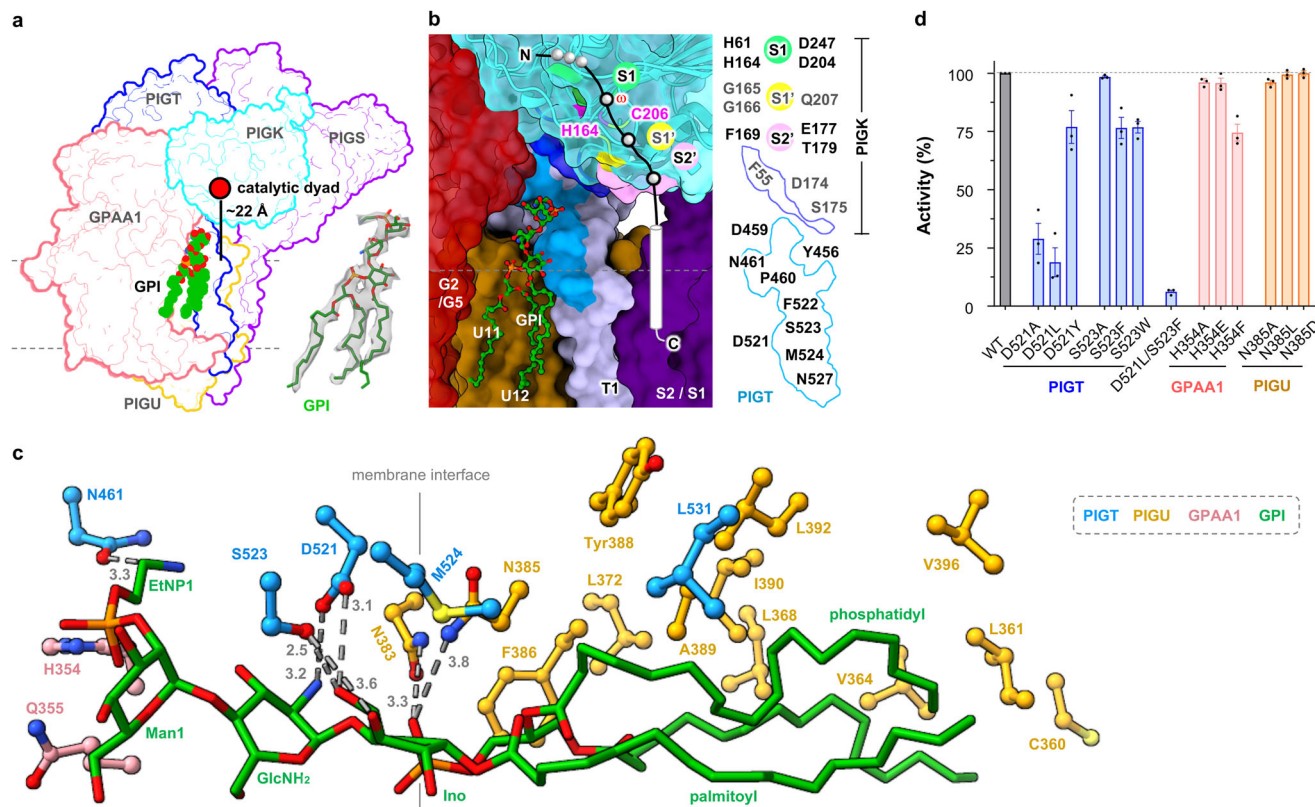

**Fig. 3 Characterization of a composite GPI-binding site. a** Cryo-EM density that fills a nearly complete GPI was observed in the membrane cavity "underneath" the catalytic dyad. **b** Expanded view of the composite site encompassed by the indicated TMHs from GPAA1 (G) /PIGU (U) /PIGT (T) /PIGS (S). Evolutionarily conserved residues (Supplementary Fig. 6c, d) in the vicinity are colored marine (PIGT) and blue (PIGK). Subunits are shown as surfaces except that PIGK was additionally shown as ribbon representations with the catalytic dyad and S1/S1′/S2′ residues highlighted in indicated colors. **c** Interaction between the partial GPI (green) and GPI-T (colored-coded as indicated). Distances (Å) are either indicated by numbers for H-bonding interactions or omitted for hydrophobic interactions (within 5 Å of GPI). A vertical line marks the membrane boundary. Although the major form of the phosphatidyl moiety in mammal cells contains 1-alkyl-2-acyl-glycerol, a diacylglycerol was modeled based on the density. **d** Apparent activity of GPI-T mutants relative to the wild-type (WT). GPI-T KO cells were gated by TGP fluorescence[41] for subunit expression and analyzed for surface staining of the reporter GPI-AP (CD59) by flow cytometry. Bar graph is color-coded to match the coloring for subunits in (**b**). Data represent mean ± s.e.m. from three independent experiments. Source data are provided.

activity (6.2%, Fig. 3d). GPAA1 H354/Q355 is within 4.5 Å of the EtNP1 group of GPI. Although the alanine mutation of H354 did not change the apparent activity, substitution with phenylalanine which has a similar volume to histidine reduced activity by ~25% (Fig. 3d). The lack of substantial change in GPI-T activity for individual mutants of other residues (Supplementary Fig. 6d) might be explained by the abundance of weak multivalent interactions with GPI.

**PIGT and PIGU form a platform for complex assembly.** PIGT has been proposed to play a scaffolding role based on its stabilizing effect on other subunits in cells[26]. Consistently, our structure reveals that PIGT and PIGU form a platform for complex assembly (Fig. 4a). On one hand, two lobes of PIGT (LbT1 and LbT2) in the ER lumen pack together to form a structure that suits a scaffolding role. Specifically, ten twisted antiparallel β-strands of LbT1 stack into a "half-rib" cage that was decorated with loops and short α-helices for interaction with GPAA1/PIGK/PIGS, and LbT2 assumes a stable β-sandwich topology for interaction with GPAA1 and PIGU (Fig. 4b). As a result, 25.9% of its total surface area was buried by other subunits (Fig. 4a). On the other hand, PIGU recruits other subunits in the ER membrane through an optimal geometry that maximizes interaction surfaces within the membrane and at the luminal surface. Specifically, its 12-TMHs are arranged into two centric

rings (Fig. 4a, c). The outer ring interacts with the transmembrane domains of all other four subunits alongside the membrane plane. The inner ring, however, contains six short TMHs that do not fully transverse the membrane, creating a hydrophobic void to strategically attract the five amphipathic helices (AH1-5) (Fig. 4c). In turn, these AHs expose several acidic residues and dipole moments (Fig. 4c) to create an overall negatively charged surface for electrostatic complementation with the corresponding PIGT surface (Fig. 4d). Thus, PIGU is arranged optimally as a docking base for PIGT and together with PIGT to buttress all other subunits.

**PIGT and PIGS may position PIGK for substrate specificity.** GPI-T is a somewhat promiscuous enzyme. It does not recognize a specific peptide sequence. Rather, it processes substrates with a small ω-residue (A/C/D/G/N/S) followed by a hydrophilic spacer of 8–12 residues (~24–36 Å in the extended form) and a hydrophobic C-terminal tail (Fig. 1a). Such characteristics, although not highly stringent, provide a pattern that is seemingly explored by GPI-T for substrate specificity through the positioning of the catalytic PIGK subunit.

In detail, the position of PIGK is secured by multivalent interfaces mainly through interactions with PIGT and PIGS. The PIGK protease domain sits snugly in a three-sided cavity formed by the PIGT/PIGU platform as well as GPAA1 and PIGS

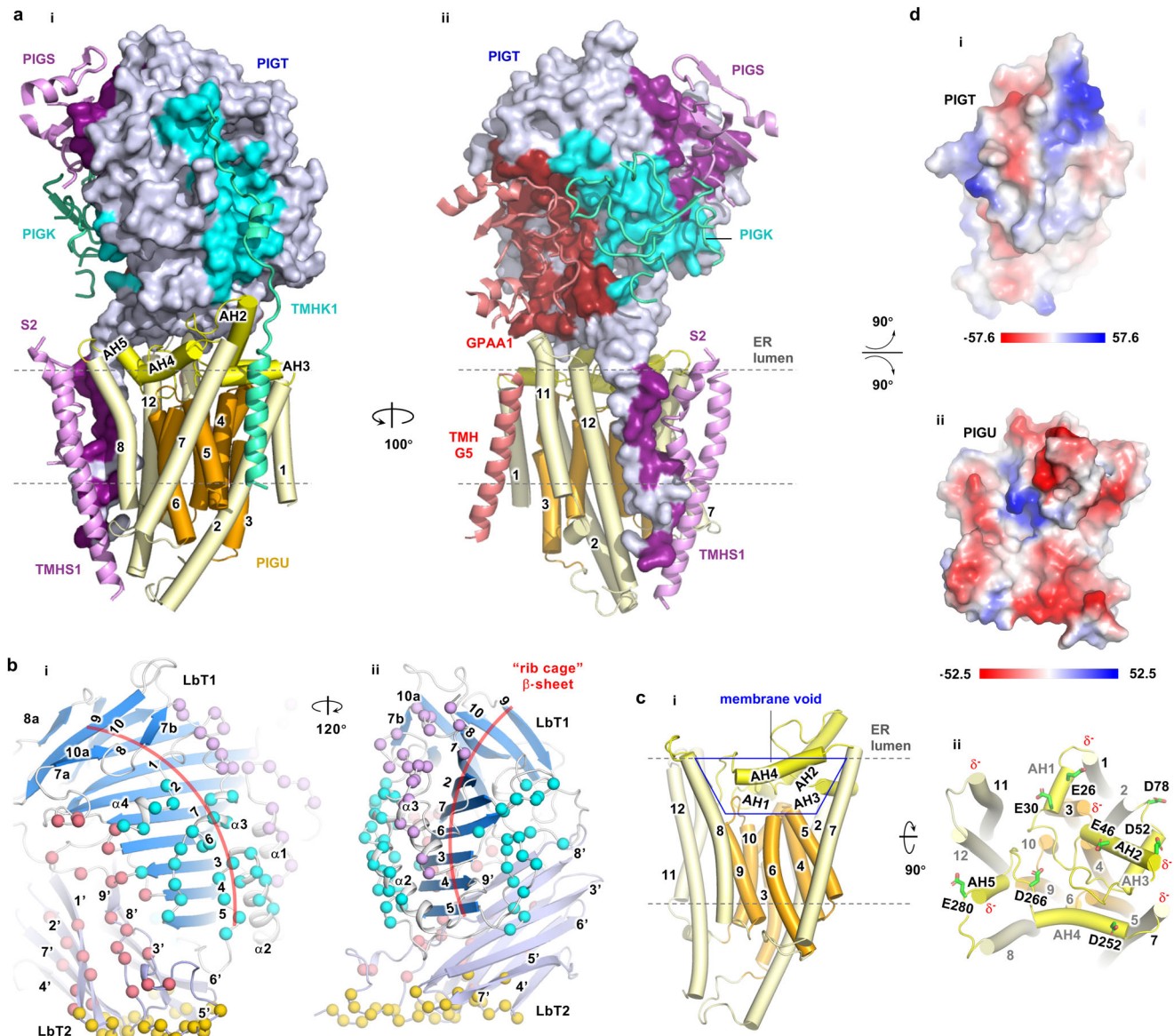

**Fig. 4 PIGT and PIGU form a platform for complex assembly. a** PIGT (surface representations, blue) and PIGU (cylinder, yellow/orange) form a docking platform for other subunits (GPAA1, red; PIGS, purple; PIGK, cyan, ribbon). Interaction surfaces on PIGT are shaded to match the color of GPAA1/PIGK/PIGS. **b** PIGT displays skeleton features. Its luminal domain contains two lobes (LbT1/2). LbT1 (marine) consists of a central "rib-cage" β-sheet (red curve, β1-10) sandwiched by connecting loops, α-helices (α1-4), and β-strands (β7a, 7b, 8a, and 10a). LbT2 (light blue) consists of two layers of β-sheets with a total of 9 β-strands (β1'-9'). Spheres indicate residues that interact with GPAA1 (red), PIGK (cyan), PIGS (purple), and PIGU (orange). **c** Side (**i**) and normal (**ii**) view of PIGU. PIGU features a membrane core region (orange) with short transmembrane helices (TMHs) enclosed by a ring of TMHs (pale yellow). This arrangement creates a membrane void (blue trapezoid) to hold the five amphipathic helixes (AH) 1-5 (yellow). The TMHs and AHs are so arranged such that the C-terminal ends of several α-helices (labeled with black text) expose to the surface (**ii**). The resulting dipole moments (δ⁻) and acidic residues (stick representation, green) make the surface electrostatically negative. **d** "Open-book" representation of the electrostatic potential molecular surface (red, negative; blue, positive; white, neutral) generated using the Adaptive Poisson-Boltzmann Solver module in PyMOL (version 2.3.3).

approaching from opposite directions (Fig. 1c). In addition, PIGK exploits two grooves on the shamrock-like PIGS for interactions, with the globular luminal domain complementing the concave Groove1 surface and its "belt"-like loop entangling onto the Groove2 (Fig. 5a). This configuration buries a total of 2341.9 Å² surface area and holds PIGK tightly via 18 hydrogen bonds, 6 salt bridges, and several hydrophobic interactions (Fig. 5b). Further, the C-terminal portion of PIGK travels down along a shallow surface of PIGT's backside and buckles its terminal helix into the membrane to interact with TMH5/7 of PIGU (Figs. 1c, 4a).

Finally, an inter-subunit disulfide bond between PIGK C92 and PIGT C182, which has been reported previously[25] and visualized in this study, nails PIGK onto PIGT (Fig. 5c, d). In line with the multivalent nature of the interactions, disruption of the disulfide bond by PIGK C92A caused a significant (~40%), but not a complete loss of the apparent GPI-T activity (Fig. 5e). These interactions place PIGK at a "mid-air" position (relative to the membrane "ground") with its catalytic dyad measuring ~22 Å from the membrane (Fig. 5c). Because the distance approximates the hydrophilic "stem" of both GPI and proprotein substrates that

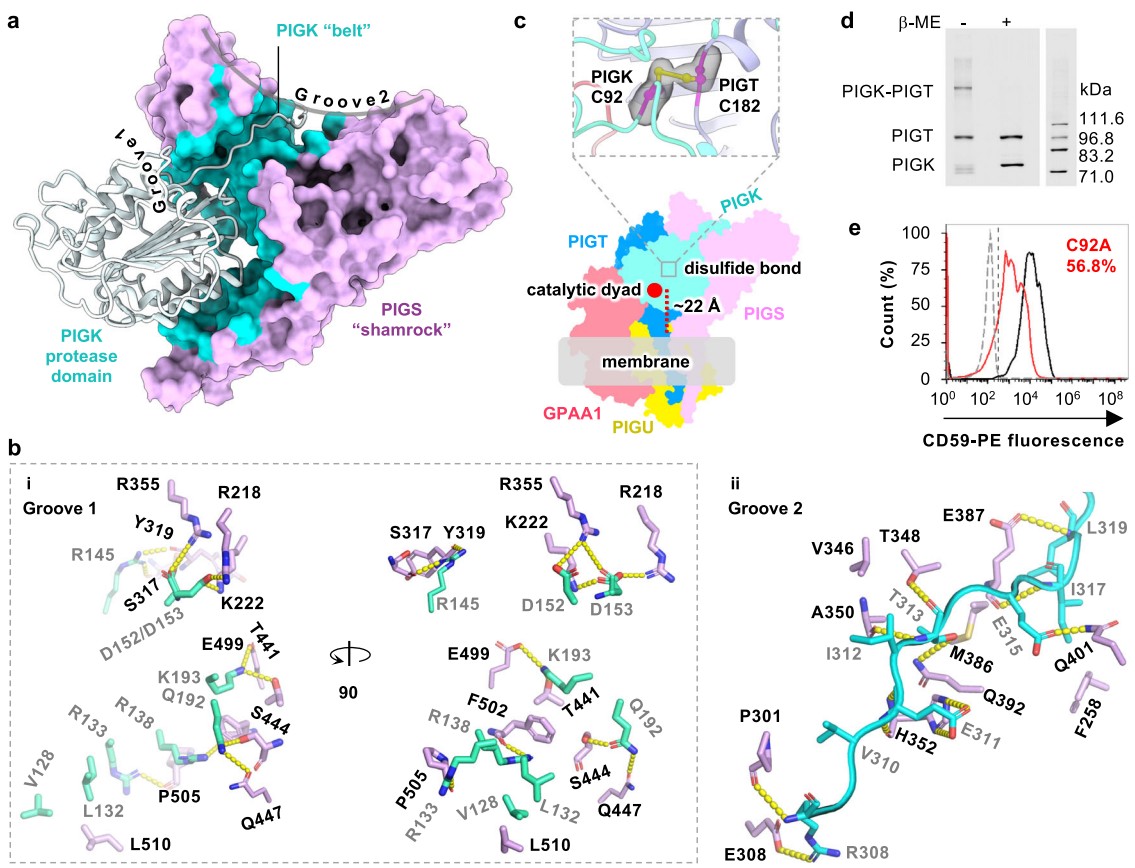

**Fig. 5 PIGS and PIGT place PIGK and its catalytic dyad in a position suitable for catalysis. a** PIGS (surface, pink) holds PIGK (ribbon) by embracing the protease domain using one of the shamrock grooves and hosting the loop region in another groove. The buried surface is colored cyan. **b** Detailed interactions between PIGK and PIGS at Groove 1 (**i**) and Groove 2 (**ii**). PIGK residues (cyan) are indicated with gray texts and PIGS residues (light purple) are labeled with black texts. Dashed lines (yellow) indicate distances within 3.6 Å. **c** An inter-subunit disulfide bond (PIGT C182 / PIGK C92) (magenta) nails PIGK (cyan) onto the PIGT (blue) (top). This and the other interactions fix PIGK in a position such that the catalytic dyad (red dot) is ~22 Å "above" the membrane interface (gray shading). This geometry would suit interactions with GPI which is expected to insert into the membrane by the phosphatidyl moiety with its hydrophilic glycan chains (measuring ~25 Å in the extended form) stemming from the membrane to meet with the catalytic dyad. The cryo-EM density for the disulfide bond is shown gray. A GPI-T structure (bottom) color-coded as indicated shows the approximate position of the disulfide bond. **d** Verification of the PIGK-PIGT disulfide bond. SDS-PAGE in-gel fluorescence shows a high-molecular-weight band (PIGK-PIGT) in the absence of, but not in the presence of, the reducing agent β-mercaptoethanol (β-ME) at the cost of the individual PIGT/PIGK bands. Free PIGT/PIGK bands under non-reducing conditions were probably from uncomplexed PIGT/PIGK proteins due to the uneven expression level of all the five subunits. Other subunits were also co-transfected but were invisible owing to their lack of the TGP-tag[41]. Theoretical molecular weights of home-made fluorescent molecular markers[69] are labeled on the right. Shown is a representative result of three independent experiments. The uncropped image is available in the Source Data file. **e** Disruption of the PIGT-PIGK disulfide bond by PIGK C92A causes loss of GPI-T activity. PIGK knockout cells expressing TGP-tagged wild-type (black), C92A (red), or a control membrane protein (gray) were gated by TGP fluorescence (for the expression of PIGK), and the sub-population was analyzed for surface staining of the reporter GPI-AP (CD59) by flow cytometry using phycoerythrin (PE)-conjugated antibodies. C92A showed an apparent activity of 61.0 ± 6.3% (s.e.m., $n = 3$) compared to the wild-type PIGK. A vertical line indicates the threshold for CD59 fluorescence. Shown is a representative result from three independent experiments. Source data are provided.

are presumably rooted in the membrane (Fig. 1a), this topological arrangement may confer substrate specificity.

**GPI-T contains two protease-like domains.** GPAA1 is the first GPI-T subunit to be discovered[47] but its functional role has remained under debate. Bioinformatic studies[23,48] have suggested a protease fold and hence a catalytic role. In our structure, GPAA1 assumes a portico shape with eight TMHs, four AHs, and a soluble domain (Fig. 6a). Whilst the transmembrane domain shares no recognizable structural homology with known folds, the soluble domain is indeed similarly arranged as the $Zn^{2+}$-protease AM-1[49] (PDB ID 2EK8 https://doi.org/10.2210/pdb2EK8/pdb, Cα RMSD of 3.2 Å) (Fig. 6b). However, a closer inspection shows that the corresponding GPAA1 domain lacks a characteristic catalytic

zinc-binding motif that consists of glutamates, aspartates, and histidines (Fig. 6c), and mutations of possible Zn-binding motif candidates in the vicinity (D153A, E186A, H187A, D188A, E226A, H303A) (Fig. 6d) had no noticeable effect on GPI-T activity in cells (Fig. 6e). These results suggest that GPAA1, even if participating in catalysis, may not use a mechanism of the homologous $Zn^{2+}$-proteases.

Likewise, a portion of PIGS (resi. 222-398) folds similarly to a metzincin-type protease AmzA (Cα RMSD of 3.6 Å)[50] (Supplementary Fig. 7a). However, unlike AmzA, PIGS lacks elements for a metzincin active site that contains a conserved methionine near a zinc-binding tri-histidine motif. Further, a cysteine-based zinc-finger in AmzA was compositionally impossible in PIGS (Supplementary Fig. 7b, c). Thus, we envisage that the protease-like GPAA1 and PIGS subunits may help recruit

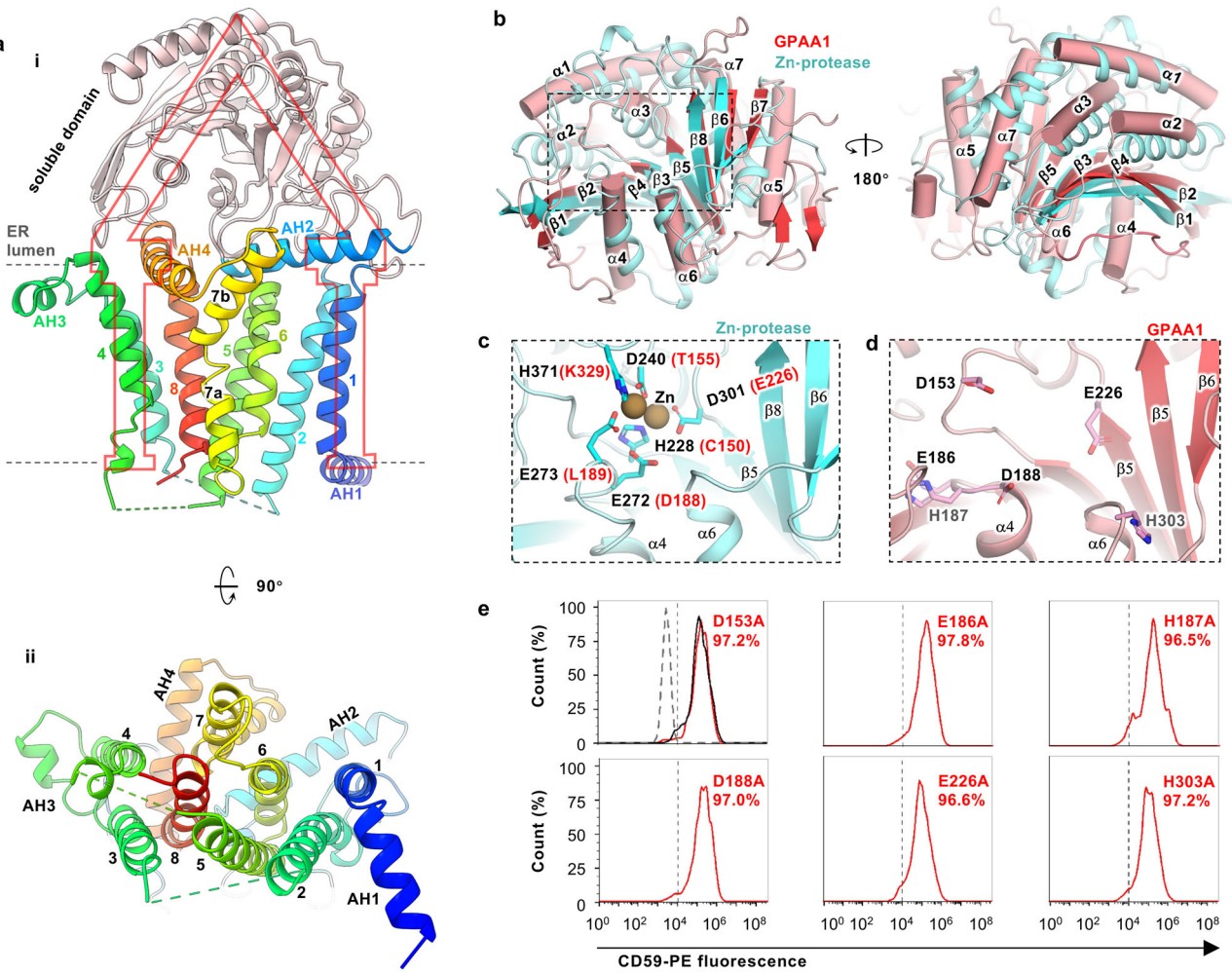

**Fig. 6 Structural and functional characterization of GPAA1 reveals a protease-like domain. a** Side (**i**) and normal (**ii**) view of GPAA1. Numbers indicate transmembrane helices (TMHs) and AH1-3 label the three amphipathic helices (AH). The soluble domain is colored pink and the membrane-associated domain is rainbow-colored (blue, N-terminal; red, C-terminal). **b** The soluble domain of GPAA1 (red/pink, cylinder) is structurally similar to a Zn-protease AM-1 (cyan, cartoon, PDB ID 2EK8 https://doi.org/10.2210/pdb2EK8/pdb) with a Z-score of 20.6 and Cα RMSD of 3.2 Å (from a DALI search)[49]. **c** The Zn-binding site of AM-1 (expanded view of the boxed region in **b**) consists of two each of aspartate, glutamate, and histidine residues that are not fully conserved in GPAA1 (in brackets). **d** Arrangement of GPAA1 aspartate/glutamate/histidine (D/E/H) residues in the region corresponding to the Zn-binding site in AM-1. Despite having the same composition, these residues are unlikely to form a Zn-binding site because of different spatial arrangements, especially for the two histidine residues (gray text). **e** Mutation of the D/E/H residues in (**d**) did not reduce CD59 staining in the flow cytometry assay. The function of wild-type GPAA1 and mutants were assessed by the surface expression of the GPI-AP reporter (CD59) in GPAA1 knockout cells transfected with appropriate plasmids. Cells were gated by TGP fluorescence[41] for GPAA1 expression and analyzed for CD59 staining using phycoerythrin (PE)-conjugated antibodies. The dotted line (gray) indicates the CD59-staining background level from cells expressing an unrelated TGP-tagged membrane protein (negative control). Solid lines indicate staining of cells transfected with the wild-type (black) or mutant genes (red). A vertical dash line marks the threshold (CD59-gating) determined from the negative control. Shown is a representative result from three independent experiments. Data for all the three experiments are included in the Source Data file.

protein substrates instead of performing cleavage, reminiscent of the γ-secretase[51].

**Structural interpretation of pathogenic mutations**. Highlighting the indispensable role of GPI-T in GPI-AP biogenesis, malfunction of GPI-T by genetic mutations causes reduced surface expression and neurodevelopmental disorders such as NEDHCAS[28–37]. Our structure provides a framework to propose possible mechanisms for the pathophysiology of such mutations. Thus, mapping these mutations onto GPI-T showed a pattern with over half of them (12 out of 22, PIGK S53F/L86P/A87V/D88N/Y106S/A184V/M246K/C275R, PIGT V528M, GPAA1 S51L/A389P, and PIGU N383K) clustering near the catalytic dyad and the GPI-binding site (Fig. 7, Supplementary Fig. 8a). Among

the rest of the ten mutations, eight are located at the inter-subunit interfaces (PIGT T183P/E184K/G360V/G366W/R488W, PIGS E308G, GPAA1 W176S, and PIGU I70K) (Fig. 7, Supplementary Fig. 8b) and hence may disrupt the integrity of the complex. PIGS L34P occurs in the first transmembrane helix and may cause folding issues by main chain distortion. PIGK Y160S and GPAA1 W176S may also promote misfolding by introducing the small hydrophilic serine residue into a hydrophobic microenvironment (Supplementary Fig. 8c, d).

**Discussion**

As a key enzyme catalyzing the committed step in the GPI-AP biogenesis pathway, GPI-T has been extensively studied and a 3D structure has been long sought for the mechanistic understanding

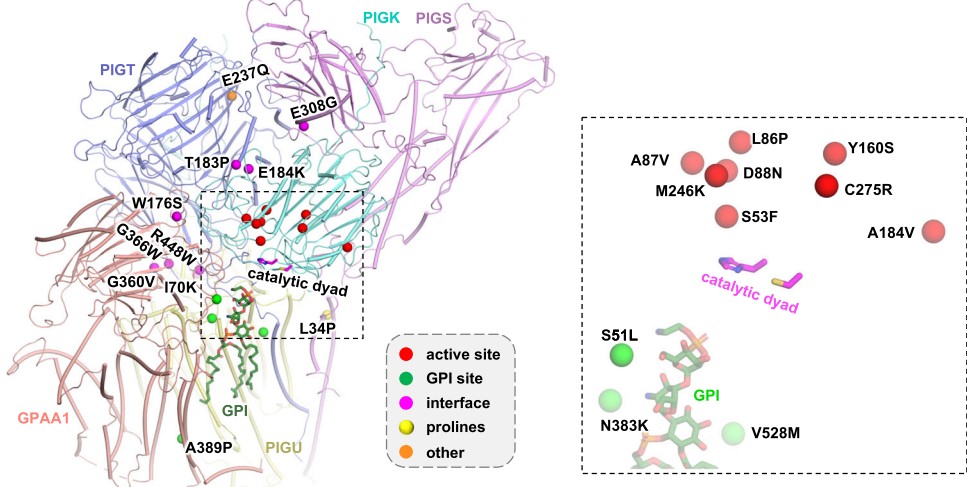

**Fig. 7 Distribution of genetic mutations on GPI-T.** Mapping the disease mutations (Cα spheres, color-coded by the categories in the gray box) onto the human GPI-T structure (main chain, color-coded cylinder representations; catalytic dyad, magenta stick presentations; GPI, green stick representations). The mutations include PIGK S53F/L86P/A87V/D88N/Y160S/A184V/M246K/C275R that cause a neurodevelopmental syndrome with hypotonia, cerebellar atrophy, and epilepsy[28], PIGT T183P that causes an intellectual disability syndrome[31], PIGT E184K/G360V/R448W that are related to the Multiple Congenital Anomalies-Hypotonia Seizures Syndrome 3[32, 35, 36], PIGT E237Q/V528M that cause developmental disorders characterizing learning disability, epilepsy, microcephaly, congenital malformations and mild dysmorphic features[33], PIGT G366W that is found in patients with epileptic apnea and multiple congenital anomalies, severe intellectual disability, and seizures[34], PIGS L34P/E308G that are related to a neurological syndrome ranging from fetal akinesia to epileptic encephalopathy[30], PIGU I70K/N383K that are found in patients with severe intellectual disability, epilepsy, and brain anomalies[29], and GPAA1 S51L/W176S/A389P that are also related to developmental disorders featuring global developmental delay, epilepsy, cerebellar atrophy, and osteopenia[37]. Left, overview with a box showing the area close to the active site. Right, the expanded view of the boxed region on the left.

of its assembly and catalysis. In this study, we report the 2.53Å-resolution structure of the human GPI-T complex with an endogenous GPI molecule. Combined with rational mutagenesis, the structure reveals an unexpected composite GPI-binding site within which a juxta-membrane portion of PIGT is found to contribute most to the interactions. Mutagenesis also identified critical residues near the catalytic dyad as potential determinants for proprotein substrate binding with a similar mechanism to that of legumains. Structural analysis suggests an assembly mechanism whereby GPI-T explores the hydrophobicity/hydrophilicity pattern of the substrates for specificity through a geometry where the distance between the catalytic dyad and the membrane interface acts as a molecular ruler to filter proprotein and GPI substrates.

Although the substrate binding and catalytic mechanisms are yet to be revealed by structural studies of GPI-T with the peptide substrate, the structural comparison to legumains offers mechanistic implications. In legumains, the trivalent oxyanion pocket[44] (backbone amide nitrogen of C189, G149, and Nδ1 of the catalytic residue H148) is proposed to polarize the carbonyl oxygen of the P1 residue in the substrate. This increases the electrophilicity of the carbonyl carbon, allowing the catalytic C189-Sγ to deprotonate and launch the nucleophilic attack that proceeds to peptide cleavage. In PIGK, the corresponding PIGK residues are the similarly positioned C206, G165, and H164 (Supplementary Fig. 4d, e), indicating an analogy in the carbonyl activation process. In legumains and caspases, the selectivity of the substrate P1 residue is conferred mainly by the S1 pocket, and the characteristic zwitterionic S1 site (two positively charged residues and two negatively charged residues) of legumains is believed to be suited for its asparagine specificity. In line with this, legumains are unable to process substrates with an aspartate unless low-pH conditions are used to protonate the P1 residue[44]. Interestingly, despite being capable of processing substrates with various ω-residues (analogous to P1) including aspartate, the corresponding site in PIGK has virtually the same composition as

the S1 of legumains except for a conservative change (PIGK D204 versus legumain E187) (Fig. 2a, Supplementary Fig. 4e). Whether and how this site confers selectivity to proprotein substrates remains to be investigated structurally and biochemically.

In addition to the protease activity, a ligation reaction to conjugate the GPI-peptide amide, presumably also by PIGK, is required for GPI anchoring. In legumains[52], the ligase activity involves a recently discovered mechanism where a succinimide residue converted from D147 provides energy. Despite its overall similarity with legumains, PIGK contains a glycine (G163) as a counterpart to the legumain D147 (Supplementary Fig. 4a), thus making the succinimide-mediated ligation less likely in PIGK. Instead, PIGK may use a ping-pong mechanism similar to that of the sortase transamidation[53]: the catalytic dyad acts on the proprotein, forming a semi-stable thioacyl intermediate that links the ω residue and C206. The incoming amine group on EtNP3 then attacks this intermediate, producing GPI-AP and freeing PIGK.

GPAA1 has been proposed previously to be a catalytic subunit owing to its similar arrangement of secondary structure elements to that of zinc proteases[23]. We verify the structural similarity but clarify that GPAA1 lacks the Zn-binding motif (Fig. 6b–e). Furthermore, we would note that the region corresponding to the AM-1 Zn-binding site is remote (~50 Å) from the catalytic dyad in our structure (Supplementary Fig. 9a).

Both GPAA1 and PIGU have been proposed to be involved in GPI-binding. The functional part was narrowed down to the last TMH for GPAA1[24] which is wrapped by the rest of the GPAA1 TMHs (Fig. 6a). Because this site is ~20 Å away from the composite GPI-binding site (Supplementary Fig. 9a), how GPAA1 TMH8 is involved in GPI-binding is currently unclear. Similarly, residues F274/W275, located in the AH5 of PIGU (Supplementary Figs. 2c, 9b), are proposed to be crucial for GPI-binding[20]. Although the functional importance of the two residues is verified in this study (Supplementary Fig. 9c), they are also distal (~10 Å) to the composite GPI-binding site (Supplementary Fig. 9b).

Mutations in these regions may compromise GPI-binding by allosteric effects.

The density for a GPI molecule in the structure provides valuable information for an overview of the active site, but the detailed mechanism, for example, the requirement of the EtNP3 for GPI-T reaction, remains unknown because of the lack of reliable density beyond Man1. In addition to improving resolution, designing mutants to covalently trap GPI or GPI-APs to the active site should help reveal the complete picture of substrate binding.

The glycan-containing digitonin molecule proximal to the composite GPI-binding site may have functional implications. Because lipids mostly diffuse two-dimensionally in the membrane plane, GPI-T may use the cavity as a "hamster" pocket to stuff GPI. By doing so, the relatively low diffusion rate of lipids may be compensated by the locally high concentration of substrates in a reaction.

The interpretation of the apparent activity from the GPI-AP reporter assay is worth discussing. Assays for wild-type and mutants should ideally be performed under conditions where activity responds linearly with the enzyme loading for normalization reasons. In cell-based assays where concentrations of enzyme and substrates are less controllable, the actual activity could be either over-estimated or under-estimated. Over-estimation could happen if the expression of a mutant is higher than the wild-type. However, this limitation is less problematic for a complex than for single-component enzymes. Unlike single-component enzymes, overexpression of any individual subunits at the endogenous background level of other subunits would not increase the functional concentration of the complex (unless the subunit increases the level of other subunits by transcriptional/translational regulation or stability reasons). Thus, the change of apparent activity should be attributed to the mutation of interest, except for mutations that compromise complex assembly in which case the defects could be under-evaluated owing to the possible concentration-dependent promotion of complex formation. On the other hand, underestimation could happen if the expression of a mutant is lower than the wild-type. While it is true that not all mutants were expressed at equal levels and hence this scenario remains possible, we note that the staining of the GPI-AP reporter was counted only for the TGP-positive cells (as an indication of the expression of the subunit of interest). Although some TGP-positive signals may be due to free-TGP from degradation, a cleaved C-terminal TGP does not necessarily translate to a degraded and nonfunctional subunit. Further, we found that the levels of degradation were minor, and were comparable between the wild-type and mutants for particular subunits according to the in-gel fluorescence results (Supplementary Fig. 10). Finally, we should note that the reasons for loss-of-function mutants for the PIGK and GPI-binding sites were interpreted as being critical for substrate binding, but the possibility exists that they may compromise activity by affecting inter-subunit interactions, especially for those at the shared cavity.

GPI-T is an evolutionarily conserved enzyme and the yeast homolog has been reported to exist in dimer based on native PAGE results[54]. Although we have not observed dimer particles for the human GPI-T in this study, we note the report of an increasing number of detergent-specific weak oligomers of membrane proteins in the literature[55–58]. Whether GPI-T function as the current heteropentamer or a higher-order dimer remains to be investigated in the future. Such studies, and in fact enzyme kinetics studies, would greatly benefit from a robust test-tube biochemical assay with authentic substrates which is yet to be developed. For the same reason, it remains to be investigated if our method of GPI-T purification preserves its enzymatic activity. However, the fact that the complex showed a Gaussian peak on gel filtration, the enzyme's ability to trap an endogenous GPI ligand at its active site after lengthy solubilization/detergent exchange/chromatography steps, and the consistency between the structural observation and previous biochemical knowledge of the complex suggest functional relevance of our structure.

During the submission of our manuscript, Zhang et al. published the structure of the GPI-T solubilized in the glyco-diosgenin detergent at 3.10-Å resolution (PDB ID 7W72 https://doi.org/10.2210/pdb7W72/pdb)[59] with consistent results reported in this work. The two structures are overall highly similar with a Cα RMSD of 0.997 Å. Compared with 7W72 which identified only one phospholipid (GPI), the higher-resolution map from this work allows the visualization of 22 lipid/detergent molecules (Fig. 1c) and more details of the GPI ligand in both the acyl chain and the glycan core (Supplementary Fig. 11). The 7W72 model adopts a slightly more compact build, and the catalytic dyad is ~3 Å closer to the lipid substrate (Supplementary Fig. 11). The functional significance of the differences remains to be investigated. It may be that the two structures represent different conformation states during the catalysis cycle such as substrate binding and product release. Alternatively, the distance between the catalytic dyad and membrane interface needs to have some flexibility to accommodate proprotein substrates with different lengths in the spacer region of the CSP (Fig. 1a).

In summary, our work formulates feasible working models for the GPI anchoring process and provides a high-resolution framework to inspire the design of biochemical and biophysical experiments to probe catalytic mechanisms in detail.

## Methods

**Molecular cloning.** The genes encoding the human GPI-T subunits GPAA1 (Genbank ID NP_003792.1), PIGK (NP_005473.1), PIGS (NP_149975.1), and PIGT (NP_057021.2) were PCR amplified from cDNA clones provided by the authors' institute. The gene encoding the human PIGU (NP_536724.1) was PCR amplified using overlapping oligonucleotides. The PCR products were Gibson assembled[60] into various versions of the pBTSG[41] vector which were modified to carry the following tag sequences at the 3'-end of the encoding sequence of the thermostable fluorescence protein (TGP)[41] tag: GPAA1, 2×Flag; PIGK, hemagglutinin (HA); PIGS, Myc; PIGT, 9×His; PIGU, Strep. For disulfide cross-linking between PIGT and PIGK, the DNA fragment encoding TGP was removed in the constructs of GPAA1/PIGS/PIGU to avoid background in in-gel fluorescence. The constructs were verified by DNA sequencing.

Mutations were made using standard PCR-based site-directed mutagenesis. DNA sequences were verified by Sanger sequencing.

The TGP tags are located on the cytosol side of the GPI-T, opposite to its large luminal domain. Flow cytometry (below) showed that the TGP-tagged GPI-T was functional because the co-transfection of all five TGP-tagged subunits in a cell line with all GPI-T subunits disrupted (see below) restored cell surface expression of the GPI-AP reporter CD59.

**Generation of GPI-T knockout (KO) cell lines.** To generate cells lines with defective individual subunits, the endogenous genes encoding the five GPI-T subunits were separately disrupted by CRISPR-Cas9 editing using two or three pairs of sgRNA oligos that are designed using the online server (http://cistrome.org/SSC/)[61] with the following sequences (forward / reverse pairs):
5'- CACCGTGTGGGGCTGCTGGCAC-3'/5'- AAACGTGCCAGCAGCAG CCCCACAC-3' and 5'- CACCGCAGGAGCAAGAAGCCGACAG-3'/5'- AA AAC CTGTCGGCTTCTTGCTCCTGC-3' for GPAA1; 5'- CACCGAATTACCAA CATA GAACTCG -3'/5'- AAACCGAGTTCTATGTTGGTAATTC -3', 5'- CAC CGTTC ATATTAGTTTGGCTAGC -3'/ 5'- AAACGCTAGCCAAACTAATATG AAC -3' and 5'- CACCGGCTCTAGCTAGTAGTCAAGT -3'/5'- AAACACTTGA CTACTAGCT AGAGCC -3' for PIGK; 5'- CACCGTGAGCCTCAGGAACAAG CGG -3'/5'- AAA CCCGCTTGTTCCTGAGGCTCAC -3' and 5'- CACCGAGTG GAGCGCTGAGAA GAGG-3'/5'- AAACCCTCTTCTCAGCGCTCCACTC-3' for PIGS; 5'-CACCGC GGTGCAGACCACCTCCCG -3'/5'- AAACCGGGAGGTGG TCTGCACCGC-3' and 5'- CACCGCACCATCACCTCCAAGGGCA-3'/5'- AAA CTGCCCTTGGA GGTGATGGTGC-3' for PIGT; 5'- CACCGAGTCCTGGAT TGCAAAATAC-3'/5'- AAACGTATTTTGCAATCCAGGACTC -3', 5'- CACCGC CTAATTGACTATGCT GAAT-3'/5'- AAACATTCAGCATAGTCAATTAGGC -3' and 5'- CACCGTCTTT GGGTAGTCAAAGTGA-3'/5'- AAACTCACTTTG ACTACCCAAAGAC-3' for PIGU.

The oligos were designed to have sticky ends that are compatible with the Type IIs restriction enzyme BbsI (Cat. R3539S, NEB) after annealing. The sgRNA oligo

pairs, dissolved in a buffer containing 0.2 M NaCl, 0.1 mM EDTA, and 10 mM Tris HCl pH 7.5, were mixed at equimolar concentrations of 10 μM in a PCR tube. The oligonucleotides were first denatured by heating at 95 °C for 3 min, before being subjected to an annealing step with gradual cooling from 94 °C to 25 °C at 1 °C gradients and an 11-s incubation under each temperature. One microliter of the annealed mix was ligated into the vector pX330 (50 ng) pre-digested with *Bbs*I using 5 units of T4 ligase (Cat. EL0011, Thermo Fisher Scientific) in a 10-μL reaction system at room temperature (RT, 20–22 °C). The ligation products were transformed into DH5α and the resulting colonies were sent for DNA sequencing to identify desired constructs.

For CRISPR-Cas9 gene editing, 8 μg of sgRNA-encoding plasmids constructed above, 0.16 μg of pMaxGFP (as a FACS marker later), 16 μL of P3000 (Cat. L3000008, Thermo Fisher Scientific), and 250 μL of Opti-MEM medium (Cat. 31985070, Thermo Fisher Scientific) were mixed and added to a separately prepared mix containing 16 μL of Lipofectamine 3000 and 250 μL of Opti-MEM medium. After incubation at RT for 15 min, the mixture was added dropwise to a 6-cm dish containing HEK293 cells with 70–90% confluency. Cells were cultured in a 5% CO$_2$ incubator at 37 °C in a Dulbecco's Modified Eagle Medium (DMEM) supplemented with 10% fetal bovine serum (FBS). After 24 h, cells were washed with 2 mL of PBS and digested with 0.5 mL of 0.1% trypsin (Cat. 25200056, Thermo Fisher Scientific) for 3 min at 37 °C before being resuspended in 3 mL of DMEM and 10% FBS to saturate trypsin. Cells were then harvested by centrifugation at RT at 300 g for 5 min, washed with 10 mL of PBS, and resuspended in 0.5 mL PBS for fluorescence assisted cell sorting (FACS). A total of ~800,000 cells were sorted in a BD FACSAria Fusion machine with software BD FACS Diva (version 8.0.3) by green fluorescence protein (GFP) and the top 5% (~40,000) were collected. These cells were serially diluted using DMEM supplemented with 10% FBS and seeded into 96-well plates such that an average of 2 or 4 cells were contained in each well with 100 μL of medium. Single cells were allowed to populate for 10–12 d in a stationary CO$_2$ incubator. Wells containing a single colony were selected under a microscope, and the selected cells were washed with 30 μL pBS before being treated with 30 μL of 0.1% trypsin for 2 min at 37 °C. After this, cells were resuspended in 200 μL of DMEM and 10% FBS and divided into two parts (160 μL and 70 μL) for further culturing and PCR-identification, respectively. The alternation of genome was screened using the following primer pairs: GPAA1, 5′-AGGACTCCGGGTTTAGGT CT-3′ and 5′-GTAGCCCAATC AAGGACCCC-3′; PIGK, 5′- TAAGCGATCTGC CCTACCAC-3′ and 5′-CCCAC AGGGAAGAATTC GGG-3′; PIGS, 5′-GGCGAA ATGGGTGTCATGTG-3′ and 5′-GGCATGCAGATT TCCCTCCT-3′; PIGT, 5′-GACTGTGCTTAAGGAGGGC A-3′ and 5′-AGCCTAA CGTTGCCAAACCC-3′; PIGU, 5′-GCACAAAATGGTC CGGCAG-3′ and 5′- AGGCCCATTAAGGCCAAG TT-3′. Cells pellets were resuspended into a mix containing 10 μL of 0.1 mM EDTA, 1% Tween-20, 1× Taq polymerase buffer (50 mM KCl, 2 mM MgCl$_2$, 20 mM Tris HCl pH 8.4), and 1 mg mL$^{-1}$ proteinase K. The mixture was heated successively at 56 °C for 2 h and 95 °C for 30 min in a thermocycler, before PCR amplification for 40 cycles. Compared to the wild-type cells, PCR products from knockout (KO) cells lacked a large band (GPAA1, 2 kb; PIGS, 2.7 kb; PIGK, 3.7 kb; PIGT, 3.4 kb; PIGU, 2.3 kb) but showed a smaller band (GPAA1, 1 kb; PIGS, 0.7 kb; PIGK/PIGT/PIGU, 1.2 kb) owing to the deletion. The genomic deletions were further verified by DNA sequencing of the PCR products using the above-mentioned primers.

To confirm the lack of GPI-AP surface expression in the individual KO cells, cells identified as positive above were allowed to grow for two more passages before being analyzed by flow cytometry (see the section below). Cells that (1) lacked CD59 on cell surface, and (2) the surface expression of CD59 could be restored by ectopic expression of the corresponding subunits were either used for further flow cytometry assays or cryo-preserved in 10% DMSO, 45% FBS in DMEM for long storage.

To generate cells with all five GPI-T subunits disrupted, the procedure above was performed in tandem. Defective GPI-T in this cell line was confirmed by DNA sequencing as above, and by the lack of CD59 staining upon transfection of combination of any four, but not all five, subunits.

**Flow cytometry**. Wild-type or GPI-T KO HEK293 cells were cultured in Dulbecco's Modified Eagle Medium (DMEM) supplemented with 10% FBS at 37 °C in 24-well plates inside a CO$_2$ stationary incubator. Plasmids (0.5 μg of single plasmids, or a total of 0.5 μg of multiple plasmids) were mixed with 1 μL of P3000, 25 μL of Opti-MEM medium, and added to a separate mix containing 1.5 μL Lipofectamine 3000 and 25 μL Opti-MEM medium. Transfection was carried out as outlined above. Two days after transfection, cells were washed with PBS, treated with trypsin as in the previous section, and washed and resuspended in 0.5 mL PBS. Phycoerythrin (PE)-labeled CD59 antibody (12-0596-42, Thermo Fisher Scientific, 1: 500 dilution) was incubated with the cells for 15 min in dark. Cells were washed with PBS and resuspended in ~0.3 mL of PBS for flow cytometry (Beckman CytoFlex LX machine with software CytExpert (version 2.4.0.28)) monitored at two wavelength pairs (488/525 for GFP, 561/585 for PE). Cells (typically 40,000) were gated using the GFP channel (from expression of TGP-tagged GPI-T subunit(s)) and analyzed for positive signal for the PE channel (for surface staining of CD59) using the software FlowJo (version v10.0.7, BD Life Sciences). An example of the gating strategy can be found in Supplementary Fig. 12.

For the apparent activity of GPI-T mutants, the percentage of the immune staining of CD59 in singly KO cells transfected with the mutant was normalized to

the negative control (the same cell line transfected with the TGP-tagged Patched, an unrelated membrane protein) and the positive control (the same cell line transfected with plasmids carrying the wild-type subunit gene). The expression and integrity of all mutants (TGP-tagged) were also separately confirmed by SDS-PAGE in-gel fluorescence using a Fujifilm machine with FLA-9000 software[41] (Supplementary Fig. 10). Data reported in this work were from three independent experiments. The statistical analysis was performed using GraphPad Prism 9.0.0.

**GPI-T expression and purification**. GPI-T was expressed in Expi293 cells co-transfected with five plasmids carrying all subunits using polyethylenimine (PEI, Cat. 23966-1, Polysciences). Briefly, 0.5 L of cells (>95% viability) at a density of 2 × 10$^6$ mL$^{-1}$ were diluted 2 times and cultured at 37 °C in a 3 L flask in a CO$_2$ incubator to re-reach 2 × 10$^6$ mL$^{-1}$ (typically takes one day). Six milligrams of plasmids and 12 mg of PEI were mixed in 100 mL of medium for 20 min at RT before being added into 1 L of cell culture. Sodium valproate (Cat. P4543, Sigma) was added to a final concentration of 2 mM. Cells were harvested after 48 h by centrifugation at 1500 g for 15 min, washed once with PBS buffer, flash-frozen in liquid nitrogen, and stored at −80 °C until use.

GPI-T was purified by tandem affinity chromatography followed by gel filtration. All steps from 4 L of culture were performed at 4 °C. Cells were resuspended with solubilization buffer containing 1% lauryl maltose neopentyl glycol (LMNG)/0.1% cholesteryl hemisuccinate (CHS), 150 mM NaCl, 1 mM PMSF, 1 × protein inhibitor cocktail (Cat. B14001, Bimake), and 50 mM Tris HCl pH 8.0 and stirred gently for 2 h. Cell debris were removed by centrifuging at 4,4200 g for 1 h. The supernatant was collected, mixed with 4.5 mL pre-equilibrated Strep Tactin beads (Cat. SA053100, Smart-lifesciences) and agitated gently for 2 h. The beads were packed into a gravity column (Cat. 7321010, Bio-Rad) and washed with 5 column volume (CV) of 0.01% LMNG, 0.001% CHS and 0.1% Digitonin (Cat. D82515, ABCone) before being incubated with 0.2% digitonin in Buffer A (150 mM NaCl, 20 mM Tris HCl pH 8.0) for 1 h. The beads were then washed with 1.5 CV of 0.2% digitonin and 5 CV of 0.1% digitonin in Buffer A before being eluted with 5 mM d-desthiobiotin (Cat. Sc-294239A, Santa Cruz) and 0.1% digitonin in Buffer A. The pooled fractions were adjusted to contain 10 mM imidazole before incubated with 1 mL of Ni-NTA beads (Cat. 1018401, Qiagen) for 1 h with mild agitation. The beads were packed into a gravity column, washed with 5 CV of 10 mM imidazole, before being eluted using 0.3 M imidazole in 0.1% digitonin in Buffer A. The Pooled fractions were concentrated with a 100-kDa cut-off concentrator (Cat. UFC810096, Merck millipore) and further purified with gel filtration on a Superose 6 10/300 GL column (Cat. 29-0915-96, Cytiva) in a running buffer containing 0.1% digitonin in Buffer A. Elution profile was monitored using a Bio-Rad NGC chromatography system with software ChromLab (version 3.3.0.09). Peak fractions were pooled and concentrated to 14–20 mg mL$^{-1}$ for cryo-EM grid preparation. Protein concentration was determined by measuring A$_{280}$ using a theoretical molar extinction coefficient of 596,940 M$^{-1}$ cm$^{-1}$ assuming an equimolar stoichiometry. The quality of purified protein was checked by in-gel fluorescence and Coomassie staining SDS-PAGE (Fig. 1b, and uncropped images in Source Data).

**Cryo-EM sample preparation and data collection**. Purified GPI-T complex (2.5 μL) was applied onto glow-discharged Quantifoil Au R1.2/1.3 (300 mesh) grids, and blotted with filter paper for 3 s with a blotting force of 5 at 4 °C, with 100% humidity in a Vitrobot Mark IV (FEI) chamber before being cryo-cooled in liquid ethane.

Grids were loaded in a Titan Krios cryo-electron microscope (Thermo Fisher) operated at 300 kV with a 50-μm condenser lens aperture, spot size 5, magnification at 165,000× (corresponding to a calibrated sampling of 0.85 Å per physical pixel), and a K2 direct electron device equipped with a BioQuantum energy filter operated at 20 eV (Gatan). Movie stacks were collected automatically using EPU2 software (version 2.91 Thermo Fisher) with the K2 detector operating in counting mode at a recording rate of 5 raw frames per second and a total exposure time of 5 s, yielding 25 frames per stack and a total dose of 52.5 e$^-$/Å$^2$.

**Cryo-EM data processing**. Cryo-EM data were processed using Relion (v3.1)[62] and CryoSPARC (v3.1)[63]. The frame motion of a total of 4,705 images stacks was corrected by MotionCor2[64] wrapper in Relion. Exposure-weighted averages were then imported to CryoSPARC and the contrast transfer function parameters for each micrograph were estimated by CTFFIND4[65]. 2D classification template was generated from a small set of particles via blob-picking. A total of 2,959,791 particles were autopicked using this template and extracted with a box size of 280 pixels, and subjected to several rounds of 2D classification and heterogeneous refinement (3D classification) to remove contaminants or poor-quality particles. A set of 329,617 good GPI-T particles was obtained and converted for Bayesian polishing in Relion, which was subsequently imported back to CryoSPARC for heterogeneous refinement, enabling the production of a 2.66 Å map from 179,871 particles by non-uniform refinement (NU-refine). A further round of heterogeneous refinement (3D classification) with phase randomized reference maps (20/30/40 Å) narrowed the dataset down to 151,590 particles, which was used to generate the final 2.53 Å map via local refinement with a membrane micelle-removed mask. Resolution of these maps was estimated internally in CryoSPARC

by gold-standard Fourier shell correlation using the 0.143 criterion. Details for data processing are in Supplementary information (Supplementary Fig. 3) and Supplementary Table 1.

**Model building**. The model was ab initio built using Coot[66] (version 0.9.6) based on the amino-acid sequence and guided by the cryo-EM density using aromatic residues, glycosylation sites, and disulfide bonds as reference markers. The model was refined with Phenix. real_space_refine[67] (version 1.19.2-4158), yielding an averaged model–map correlation coefficient (CCmask) of 0.84. Structures were visualized using UCSF ChimeraX1.1[68] and PyMOL (version 2.3.3) (https://pymol.org/2/).

**Reporting summary**. Further information on research design is available in the Nature Research Reporting Summary linked to this article.

## Data availability

The data that support this study are available from the corresponding authors upon reasonable request. The plasmids and knockout cells lines generated in this study are available from the corresponding author (D.L.) upon reasonable requests. The cryo-EM density map has been deposited in the Electron Microscopy Data Bank (EMDB) with accession code EMD-32582 (GPI-transamidase complex) and the structure coordinates have been deposited in the RSCB Protein Data Bank (PDB) under the accession number 7WLD https://doi.org/10.2210/pdb7WLD/pdb (GPI-transamidase complex). Source data are provided with this paper.

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

## Acknowledgements

Cryo-EM data were collected at SKLB West China Cryo-EM Center and processed at Duyu High-Performance Computing Center in Sichuan University. We thank the staff members of the Center of Cryo-EM of Fudan University and the Cryo-EM Center at National Facility for Protein Science in Shanghai for technical support and assistance. We thank Drs. Feilong Meng, Liu Daisy Liu, and Hai Jiang, and Mr. Yin Chen for guidance on CRISPR-Cas9. This work has been supported by the National Natural Science Foundation of China (82151215, 31870726, D.L; 32171194, Q.Q.; 82041016, 32070049, Z.S.), the Strategic Priority Research Program of CAS (XDB37020204, D.L.), CAS Facility-based Open Research Program (2017, D.L.), Science and Technology Commission of Shanghai Municipality (20ZR1466700, D.L.), Ministry of Science and Technology of China (2021YFA1301900, Z.S.), and the start-up funds from Shanghai Stomatological Hospital & School of Stomatology, Fudan University (2020, Q.Q.).

## Author contributions

Y.X. established purification protocol. T.L. and Y.X. purified the complex. Y.X. constructed KO cells and performed functional assays. Y.L., Y.C., and Z.Z. prepared and screened cryo-EM grids. G.J. collected cryo-EM data under the supervision of Z.S. Z.Z. and G.J. processed cryo-EM data and produced the final map. J.B. helped with molecular cloning. D.L. initiated the project. D.L. and Q.Q. wrote the manuscript with input from Y.L., T.L., Y.X., Z.Z., and Z.S.

## Competing interests

The authors declare no competing interests.
