## [Peer Review File · Nature Communications]

Molecular insights into biogenesis of glycosylphosphatidylinositol anchor proteinsReviewers' Comments:

Reviewer #1:

Remarks to the Author:

Glycosylphosphatidylinositols (GPI) act as membrane anchors of many eukaryotic cell surface proteins. GPI is post-translationally attached to the C-terminus of the protein by a transamidation reaction, in which the C-terminal GPI attachment signal peptide in the precursor protein is replaced by a preassembled GPI glycolipid. This process is mediated by GPI transamidase (GPI-T) consisting of five membrane proteins. Xu and colleagues purified human GPI-T and determined its structure by cryoEM at 2.53Å resolution and reported a number of characteristics critical to understand GPI-anchoring mechanism. A similar study was very recently published by another group (Zhang H et al, 2022, Nat Str Mol Biol). Two papers report basically similar and consistent results, however, this report by Xu provides the structure of higher resolution and more information than the previous report did. Specifically, authors were able to locate a cavity which accommodates GPI and characterized interactions between amino acid side-chains of GPI-T subunits and a part of associated GPI, including acylated phosphatidylinositol, glucosamine and a mannose. Other parts of GPI were not clearly seen and how "bridging" ethanolamine is presented to catalytic center is yet to be determined. The paper is clearly written and data are well discussed. There is no major concern. The only point is that since there are several differences between this and the report by Zhang, it would be useful for readers if authors add discussion about the different points between two studies.

Reviewer #2:

Remarks to the Author:

In this manuscript, Xu et al, describe the structure, obtained by single particle cryo-electron microscopy (cryo-EM), of macromolecular membrane complex responsible for glycosylphosphatidylinositol (GPI) modification of proteins in the ER. The GPI transamidase complex (GPI-T) is conserved among all eukaryotes and its structure is highly significant given the fundamental role of GPI linkages in biology, and our scarce mechanistic understanding of how this covalent attachment at protein C-termini occurs. This is beautiful work, not least because of the resolution obtained (2.53Å), and the fact the maps show clear evidence of a bound GPI core. While overall, I am very appreciative of how the results are described, I have a few comments (to be intended as constructive criticisms).

1. The manuscript - how it is written and the contents - comes across as somewhat rushed. It's more descriptive than interpretative, and as such quickly becomes tedious to read. The authors should take a moment of pause, and tell the reader what the results are in the context of what their physiological significance is, and prioritize the description of their observations accordingly.
2. The bound GPI core is a major highlight (selling point) of this works. The authors should mention this clearly in the abstract, at the end of the intro, and perhaps earlier in the results section (ie rearrange sections). As presented, at the very end, it's almost lost - as an unwanted byproduct of this, it leaves this part of Fig 1 (the GPI core is present there) unexplained until the end.
3. The density of the GPI core should be shown in Fig. S3D.
4. The authors should comment on how they were able to obtain this complex with a ligand (co-purification, I assume)?
5. It would be elegant to have an independent proof of the chemical nature of the ligand. Could the authors consider MS? This is not a major/essential point given the quality of the relevant density, but it would definitely add to the quality of the work.

6. In the introduction the authors should tell the reader about the GPI linkage diversity beyond the minimal backbone. What are the differences (additional groups added, when, where, why), how (which enzymes etc) do they occur, how does this diversity relate to the structure determined here.
7. In the Results section, the authors should add a sentence or two on how and what they determined, in particular the fact that the structure is in detergent (digitonin) should be stated up front.
8. The discussion section, in my opinion, should not start with the yeast protein run on a native gel. It off-sets the attention of the reader. The authors should set the stage in this first paragraph on what they have achieved and its relevance.
9. Did the authors identify ligand-free classes of particles?
10. How do the authors explain the presence of substantial amounts of non cross-linked PIGT/K in their non-reducing SDS-PAGE gel lane? Also, the authors should show the density map for this disulfide bind. Are there mutagenesis data to accompany this structural result?
11. The (competing) work from Zhang provides the authors with an opportunity to compare the structures, and further advance our knowledge - in other words, maybe something could be learned from such a comparison. If the answer is no, the authors should at a minimum mention the other manuscript.
12. On analysis of the model, the Zn site, in my opinion, resembles more a Ca site. What is the evidence that it is indeed Zn?

Reviewer #3:

Remarks to the Author:

Summary:

Glycosylphosphatidylinositol (GPI) is a posttranslational modification found attached to the carboxyl terminus of a broad range of cell-surface proteins ranging from receptors, enzymes, and adhesion molecules. This lipid modification allows these proteins to be anchored to the extracellular face of the plasma membrane and in turn enable them to exert their roles in important processes from fertilization, neurogenesis, to immunity. A key step of GPI modification is the cleavage of the carboxyl terminal signal peptide of the precursor substrate and the covalent attachment of the EtNP moiety of GPI to the newly exposed C-terminus via a transamidation reaction. This process is catalyzed by an enzyme complex called GPI transamidase (GPI-T). This protein complex is composed of at least five proteins: PIGK, PIGT, PIGU, PIGS, and GPAA1, with PIGK thought to serve as the catalytic component. Up until very recently, little is known about the overall architecture, subunit organization, and catalytic mechanism of GPI-T. In this manuscript, Xu et al. determined the high-resolution molecular structure of this complex by cryo-EM. Their 2.53 Angstroms cryo-EM map allowed them to build a complete atomic model of human GPI-T. This structural model showed that GPI-T adopts a canon-shaped overall architecture containing 24 transmembrane helices (8 from GPAA1, two from PIGS, one from PIGT, and one from PIGK). It also revealed the protease domain of the catalytic PIGK subunit is secured within the complex through multivalent interfaces involving PIGT, PIGU, GPAA1, and PIGS. This allowed the catalytic dyad (C206-H164) of PIGK to be optimally positioned for catalysis. The authors found that PIGK structurally resembles members of the C13 cysteine protease family (which include legumains and caspases). The high-resolution structural model enabled them to clearly visualize not only the catalytic dyad, the trivalent oxyanion hole, but also the substrate binding S1, S1', and S2 pockets. To verify the importance of the residues lining the S1, S1', and S2 pockets, they generated structure-guided mutants and tested their activities using cell-based GPI-AP reporter assay. In their cryo-EM density map, they visualized a density that corresponds to an almost complete GPI core. Results from their systematic structure-guided mutagenesis carried out in conjunction with their

cell-based assays suggested that GPI-T likely engages in weak multivalent interactions with GPI. Lastly, the GPI-T structural model provided a framework to map mutations linked to different human diseases and to understand their potential impact. Overall, the authors succeeded in overcoming technical challenges to obtain high-resolution structural information of an important membrane protein complex. The structural model derived by these investigators generated insights into the overall architecture and subunit organization of GPI-T and shed light into substrate binding and catalytic mechanism of this complex. This work complements a similar structural study on the same complex that was published very recently in *Nature Structural and Molecular Biology*. It generated a structure at higher resolution that enabled a more complete visualization of the core GPI substrate. Lastly, results from the systematic mutagenesis together with cell-based activity assays uncovered the importance of residues potentially involved in substrate binding.

Comments

1. The reconstitution and purification of GPI-T complex is an important advance and will be critical to follow-up studies by other research groups. As such, the authors are advised to provide more description of this procedure the main text as well as show a SDS-PAGE gel of the purified complex in Figure 1.
2. One question that remains to be answered is whether or not the conditions used for purification would preserve the activity of human GPI-T. While developing a true in vitro GPIT activity assay is likely beyond the scope of this manuscript, it would be helpful for the authors comment briefly on why they feel the purified reconstituted complex is active and their structural data captured an active state of GPI-T.
3. The number of mutants that the authors have generated for their cell-based activity assay is very impressive. One caveat of the approach used is the authors cannot completely rule out the loss or reduction in activity is due to low expression of the mutant subunit and/or inability of the mutant subunit to be integrated into the GPI-T complex.

RESPONSE TO REVIEWER COMMENTS

Reviewer #1

Glycosylphosphatidylinositols (GPI) act as membrane anchors of many eukaryotic cell surface proteins. GPI is post-translationally attached to the C-terminus of the protein by a transamidation reaction, in which the C-terminal GPI attachment signal peptide in the precursor protein is replaced by a preassembled GPI glycolipid. This process is mediated by GPI transamidase (GPI-T) consisting of five membrane proteins. Xu and colleagues purified human GPI-T and determined its structure by cryoEM at 2.53Å resolution and reported a number of characteristics critical to understand GPI-anchoring mechanism. A similar study was very recently published by another group (Zhang H et al, 2022, Nat Str Mol Biol). Two papers report basically similar and consistent results, however, this report by Xu provides the structure of higher resolution and more information than the previous report did. Specifically, authors were able to locate a cavity which accommodates GPI and characterized interactions between amino acid side-chains of GPI-T subunits and a part of associated GPI, including acylated phosphatidylinositol, glucosamine and a mannose. Other parts of GPI were not clearly seen and how “bridging” ethanolamine is presented to catalytic center is yet to be determined. The paper is clearly written and data are well discussed. There is no major concern. The only point is that since there are several differences between this and the report by Zhang, it would be useful for readers if authors add discussion about the different points between two studies.

We thank the reviewer for the complimentary comments.

We have added a paragraph in Discussion and the **Fig. S10** to compare with the structure from Zhang et al..

“During the submission of our manuscript, Zhang *et al.* published the structure of the GPI-T solubilized in the glyco-diosgenin detergent at 3.10-Å resolution (PDB ID 7W72) with consistent results reported in this work. The two structures are overall highly similar with a C α RMSD of 0.997 Å. Compared with 7W72 which identified only one phospholipid (GPI), the higher-resolution map from this work allows the visualization of 22 lipid/detergent molecules (**Fig. 1c**) and more details of the GPI ligand in both the acyl chain and the glycan core (**Fig. S10**). The 7W72 model adopts a slightly more compact build, and the catalytic dyad is \sim 3 Å closer to the lipid substrate (**Fig. S10**). The functional significance of the differences remains to be investigated. It may be that the two structures represent different conformation states during the catalysis cycle such as substrate-binding and product release. Alternatively, the distance between the catalytic dyad and membrane interface needs to have some flexibility to accommodate proprotein substrates with different lengths in the spacer region of the CSP (**Fig. 1a**).”

Fig. S10. Comparison of GPI-T structures from Zhang *et al.* and this work. a Superposition of the recently published GPI-T structure⁶⁰ (3.10 Å, PDB ID 7W72; grey, cylinder representation) with that determined in this work (2.53 Å, PDB ID 7WLD; colored, cartoon representation). Arrows indicate local regions with position differences. **b** Expanded view of the active site of the two structures. Compared with the structure reported in this work, the catalytic dyad (H164, C206) in the 7W72 model (yellow) is ~3-Å closer toward GPI (7W72, blue; 7WLD, green). The protein part of the 7W72 model is colored light grey and its elements are labelled italic with a prime. The structure determined in this work is color coded as in **a**.

Reviewer #2

In this manuscript, Xu et al, describe the structure, obtained by single particle cryo-electron microscopy (cryo-EM), of macromolecular membrane complex responsible for glycosylphosphatidylinositol (GPI) modification of proteins in the ER. The GPI transamidase complex (GPI-T) is conserved among all eukaryotes and its structure is highly significant given the fundamental role of GPI linkages in biology, and our scarce mechanistic understanding of how this covalent attachment at protein C-termini occurs. This is beautiful work, not least because of the resolution obtained (2.53Å), and the fact the maps show clear evidence of a bound GPI core. While overall, I am very appreciative of how the results are described, I have a few comments (to be intended as constructive criticisms).

We thank the reviewer for the supportive summary and constructive criticisms for improving our work.

1. The manuscript - how it is written and the contents - comes across as somewhat rushed. It's more descriptive than interpretative, and as such quickly becomes tedious to read. The authors should take a moment of pause, and tell the reader what the results are in the context of what their physiological significance is, and prioritize the description of their observations accordingly.

We have taken this comment seriously and made efforts to incorporate functional context when describing the structural observations in the revised manuscript.

2. The bound GPI core is a major highlight (selling point) of this works. The authors should mention this clearly in the abstract, at the end of the intro, and perhaps earlier in the results section (ie rearrange sections). As presented, at the very end, it's almost lost - as an unwanted byproduct of this, it leaves this part of Fig 1 (the GPI core is present there) unexplained until the end.

Agreed.

We have revised the abstract and the introduction accordingly, and rearranged the Results section so that the GPI results are now presented in **Fig. 3** (was in **Fig. 6** in the previous version).

3. The density of the GPI core should be shown in Fig. S3D.

Agreed. The GPI core density has been added to **Fig. S3D**.

4. The authors should comment on how they were able to obtain this complex with a ligand (co-purification, I assume)?

Yes. The GPI was co-purified with the protein complex and the information has been added to the revised manuscript.

“The co-purification of the endogenous GPI ligand suggests its relatively tight binding to the complex. Indeed, in our model, the GPI core adheres to the cavity with a rich network of interactions....”

5. It would be elegant to have an independent proof of the chemical nature of the ligand. Could the authors consider MS? This is not a major/essential point given the quality of the relevant density, but it would definitely add to the quality of the work.

We agree that a Mass-spec identification would certainly strengthen the work. However, with our facilities and labs being shut down in responding to the recent Covid-19 wave in Shanghai, we'll unlikely be able to obtain publishable results in a reasonable timeframe. We hope to develop methods for the detection and analysis of the complicated GPI in the future.

6. In the introduction the authors should tell the reader about the GPI linkage diversity beyond the minimal backbone. What are the differences (additional groups added, when, where, why), how (which enzymes etc) do they occur,

We thank the reviewer for the helpful suggestion.

We have added the following paragraph and modified **Fig. S1** to explain the diversity of GPI. Because there are a lot of enzymes involved in the process, to keep the focus on GPI-T, we did not include their names in the main text but rather have included their names in the legends of **Fig. S1**.

“Structurally elucidated in the 1980s⁶, GPI lipids are bioactive⁷ and chemically diverse with a minimal backbone consisting of a phosphatidylinositol group linked to a polysaccharide core α -Man3-(1 → 2)- α -Man2-(1 → 6)- α -Man1-(1 → 4)- α -GlcN (Man1/2/3, the three mannoses; GlcN, glucosamine; italic numbers indicate carbon numbering) (**Fig. 1a, S1a**). The glycan core in the mature GPI carries additional functional groups, some of which are species- and tissue-specific. The maturation process involves multiple enzymes including mannosyl transferases and phosphorylethanolamine (EtNP) transferases (**Fig. S1b**)²⁻⁴. The EtNP on C2 of Man1 is a prerequisite for the enzymatic addition of Man3^{8,9}, after which step two EtNPs are sequentially transferred to C6 of Man3 (as the linker for GPI anchoring) and Man2 (for efficient endoplasmic reticulum-Golgi transport of some GPI-APs)². The mannosylation of Man3 at C2 is essential for GPI anchoring in some species like yeasts¹⁰. In some Trypanosoma species, the C3 of Man2, and C3/C4 of Man1 can have further saccharide decorations and the C6 of GlcN is modified with an aminoethylphosphonate (summarized in ref.¹¹) (**Fig. S1a**). Regarding the phosphatidylinositol part, an acyl chain is usually present at C2 of Ino before GPI anchoring (**Fig. S1a**) but is in most

cases (except in erythrocytes) removed immediately after GPI attachment³. Finally, the fatty chain of the phosphatidyl group also undergoes remodeling both before (**Fig. S1b**) and after the GPI-attachment step, causing acyl diversity such as diacylglycerol, alkylacylglycerol, or ceramides with varying length and unsaturation^{2-4,11}.

Fig. S1. Chemical structure of GPI and topology of GPI-T. **a** The GPI core contains a phosphatidyl inositol (blue), a palmitoyl chain (green), a glucosylamine (GlcNH₂) (cyan), and four mannoses (Man) with possible modification of ethanolamine phosphate (EtNP) on Man1/2. The reactive EtNP3 on Man3 is colored red. The numbering of relevant carbon atoms on the sugar ring is indicated by a grey number. The glycan core can be further decorated with saccharides (blue asterisk) or phosphates (black asterisk). **b** Biosynthesis of GPI. Substrates and by-product of each reaction are labelled with grey text. Enzymes catalyzing the reactions are indicated with black texts. The two steps that do not always occur are indicated with an grey arrow. A question marker denotes speculative participants. The mature GPI is used by GPI-T. Enzymatic steps carrying further modifications are not shown. The pathway cartoon was redrawn based on ref.³. Abbreviations: CoA, coenzyme A; DAG, diacylglycerol; Dol-P, dolichol phosphate; EtNP, ethanolamine phosphate; GlcNAc, N-acetylglucosamine; GlcNH₂, glucosamine; Ino, inositol; Man, mannose; PE, phosphatidylethanolamine; UDP, uridine diphosphate.

- how does this diversity relate to the structure determined here.

The chemical composition of the GPI ligand in the structure is specified in the Results section as:

“Intriguingly, densities that fit an almost complete GPI core (palmitoylated phosphatidylinositol, glucosylamine, and EtNP-modified Man1)...”

7. In the Results section, the authors should add a sentence or two on how and what they determined, in particular the fact that the structure is in detergent (digitonin) should be stated up front.

Agreed. The following text was added in the beginning of the Result section.

“To gain insights into the GPI anchoring process, we set to determine its 3D structure by starting with its recombinant expression. The human GPI-T subunits were co-expressed in HEK293 cells with a thermostable green fluorescence protein (TGP)⁴¹ tag at the C-termini to facilitate the purification process. The membrane protein complex was solubilized in the detergent lauryl maltose neopentyl glycol (LMNG) and then exchanged into digitonin on a Strep-affinity column via PIGU. A second affinity chromatography via the nonahistidine tag on PIGT, a subunit that was pre-determined to have the lowest expression level, was performed to minimize the purification of free subunits. The complex was further fractioned by gel permeation. The peak fractions contained all five subunits along with minor high-molecular-weight contaminants based on the in-gel fluorescence and Coomassie staining results (**Fig. 1b**).”

8. The discussion section, in my opinion, should not start with the yeast protein run on a native gel. It off-sets the attention of the reader. The authors should set the stage in this first paragraph on what they have achieved and its relevance.

This is indeed a helpful comment. We have included the following texts as the opening paragraph of the Discussion.

“As a key enzyme catalyzing the committed step in the GPI-AP biogenesis pathway, GPI-T has been extensively studied and a 3D structure has been long sought for the mechanistic understanding of its assembly and catalysis. In this study, we report the 2.53Å-resolution structure of the human GPI-T complex with an endogenous GPI molecule. Combined with rational mutagenesis, the structure reveals an unexpected composite GPI-binding site within which a juxta-membrane portion of PIGT is found to contribute most to the interactions. Mutagenesis also identified critical residues near the catalytic dyad as potential determinants for proprotein substrate-binding with a similar mechanism to that of legumains. Structural analysis suggests an assembly mechanism whereby GPI-T explores the hydrophobicity/hydrophilicity pattern of the substrates for specificity through a geometry where the distance between the catalytic

dyad and the membrane interface acts as a molecular ruler to filter proprotein and GPI substrates.”

9. Did the authors identify ligand-free classes of particles?

We didn't observe any stable/good particle sets showing ligand-free classes during data processing.

10. How do the authors explain the presence of substantial amounts of non cross-linked PIGT/K in their non-reducing SDS-PAGE gel lane?

An explanatory note was added to the figure legends:

“Free PIGT/PIGK bands under non-reducing conditions were probably from uncomplexed PIGT/PIGK proteins due to the uneven expression level of all the five subunits.”

- Also, the authors should show the density map for this disulfide bond.

Point taken with thanks.

- Are there mutagenesis data to accompany this structural result?

We indeed analyzed this result via mutagenesis. Disruption of the disulfide bond causes ~40% loss of apparent GPI-T activity. The result is included in **Fig. 5e**.

“In line with the multivalent nature of the interactions, disruption of the disulfide bond by PIGK C92A caused a significant (~40%), but not a complete loss of the apparent GPI-T activity (**Fig. 5e**).”

...e Disruption of the PIGT-PIGK disulfide bond by PIGK C92A causes loss of GPI-T activity. PIGK KO cells expressing TGP-tagged wild-type (black), C92A (red), or a control membrane protein (grey) were gated by TGP fluorescence, and the sub-population was analyzed for surface staining of the reporter GPI-AP (CD59) by flow cytometry. C92A showed an apparent activity of $61.0\% \pm 6.3\%$ (s.e.m., $n = 3$) compared to the wild-type PIGK. A vertical line indicates the threshold for CD59 fluorescence. Shown is a representative result of three independent experiments (**Supplementary Data 1**).

11. The (competing) work from Zhang provides the authors with an opportunity to compare the structures, and further advance our knowledge - in other words, maybe something could be learned from such a comparison. If the answer is no, the authors should at a minimum mention the other manuscript.

A paragraph (below) and **Fig. S10** have been added to the Discussion section upon request from both Reviewer #1 and Reviewer #2.

“During the submission of our manuscript, Zhang *et al.* published the structure of the GPI-T solubilized in the glyco-diosgenin detergent at 3.10-Å resolution (PDB ID 7W72) with consistent results reported in this work. The two structures are overall highly similar with a $C\alpha$ RMSD of 0.997 Å. Compared with 7W72 which identified only one phospholipid (GPI), the higher-resolution map from this work allows the visualization

of 22 lipid/detergent molecules (**Fig. 1c**) and more details of the GPI ligand in both the acyl chain and the glycan core (**Fig. S10**). The 7W72 model adopts a slightly more compact build, and the catalytic dyad is ~ 3 Å closer to the lipid substrate (**Fig. S10**). The functional significance of the differences remains to be investigated. It may be that the two structures represent different conformation states during the catalysis cycle such as substrate-binding and product release. Alternatively, the distance between the catalytic dyad and membrane interface needs to have some flexibility to accommodate proprotein substrates with different lengths in the spacer region of the CSP (**Fig. 1a**).”

Fig. S10. Comparison of GPI-T structures from Zhang *et al.* and this work. a Superposition of the recently published GPI-T structure⁶⁰ (3.10 Å, PDB ID 7W72; grey, cylinder representation) with that determined in this work (2.53 Å, PDB ID 7WLD; colored, cartoon representation). Arrows indicate local regions with position differences. **b** Expanded view of the active site of the two structures. Compared with the structure reported in this work, the catalytic dyad (H164, C206) in the 7W72 model (yellow) is ~ 3 -Å closer toward GPI (7W72, blue; 7WLD, green). The protein part of the 7W72 model is colored light grey and its elements are labelled italic with a prime. The structure determined in this work is color coded as in **a**.

12. On analysis of the model, the Zn site, in my opinion, resembles more a Ca site. What is the evidence that it is indeed Zn?

We thank the reviewer for the keen-sighted inspection of our structure. The metal ion should indeed be an Ca^{2+} because the coordination distance (2.44 Å) matches that for Ca^{2+} (2.4-2.5 Å) but not for Zn^{2+} (2.1 – 2.2 Å). We have now rectified this error in the PDB entry.

Reviewer #3

Summary:

Glycosylphosphatidylinositol (GPI) is a posttranslational modification found attached to the carboxyl terminus of a broad range of cell-surface proteins ranging from receptors, enzymes, and adhesion molecules. This lipid modification allows these proteins to be anchored to the extracellular face of the plasma membrane and in turn enable them to exert their roles in important processes from fertilization, neurogenesis, to immunity. A key step of GPI modification is the cleavage of the carboxyl terminal signal peptide of the precursor substrate and the covalent attachment of the EtNP moiety of GPI to the newly exposed C-terminus via a transamidation reaction. This process is catalyzed by an enzyme complex called GPI transamidase (GPI-T). This protein complex is composed of at least five proteins: PIGK, PIGT, PIGU, PIGS, and GPAA1, with PIGK thought to serve as the catalytic component.

Up until very recently, little is known about the overall architecture, subunit organization, and catalytic mechanism of GPI-T. In this manuscript, Xu et al. determined the high-resolution molecular structure of this complex by cryo-EM. Their 2.53 Angstroms cryo-EM map allowed them to build a complete atomic model of human GPI-T. This structural model showed that GPI-T adopts a canon-shaped overall architecture containing 24 transmembrane helices (8 from GPAA1, two from PIGS, one from PIGT, and one from PIGK). It also revealed the protease domain of the catalytic PIGK subunit is secured within the complex through multivalent interfaces involving PIGT, PIGU, GPAA1, and PIGS. This allowed the catalytic dyad (C206-H164) of PIGK to be optimally positioned for catalysis. The authors found that PIGK structurally resembles members of the C13 cysteine protease family (which include legumains and caspases). The high-resolution structural model enabled them to clearly visualize not only the catalytic dyad, the trivalent oxyanion hole, but also the substrate binding S1, S1', and S2 pockets. To verify the importance of the residues lining the S1, S1', and S2 pockets, they generated structure-guided mutants and tested their activities using cell-based GPI-AP reporter assay. In their cryo-EM density map, they visualized a density that corresponds to an almost complete GPI core. Results from their systematic structure-guided mutagenesis carried out in conjunction with their cell-based assays suggested that GPI-T likely engages in weak multivalent interactions with GPI. Lastly, the GPI-T structural model provided a framework to map mutations linked to different human diseases and to understand their potential impact.

Overall, the authors succeeded in overcoming technical challenges to obtain high-resolution structural information of an important membrane protein complex. The structural model derived by these investigators generated insights into the overall architecture and subunit organization of GPI-T and shed light into substrate binding and catalytic mechanism of this complex. This work complements a similar structural study on the same complex that was published very recently in *Nature Structural and Molecular Biology*. It generated a structure at higher resolution that enabled a more

complete visualization of the core GPI substrate. Lastly, results from the systematic mutagenesis together with cell-based activity assays uncovered the importance of residues potentially involved in substrate binding.

We thank the reviewer for the supportive summary and specific comments below to improve our manuscript.

Comments

1. The reconstitution and purification of GPI-T complex is an important advance and will be critical to follow-up studies by other research groups. As such, the authors are advised to provide more description of this procedure the main text as well as show a SDS-PAGE gel of the purified complex in Figure 1.

We have included the gel filtration profile and the SDS-PAGE results in **Fig. 1b**.

...**b** GPI-T shows a near-Gaussian peak on gel filtration and all five subunits are present on an SDS-PAGE (inset) visualized by in-gel fluorescence (left) and Coomassie staining (right). V_0 and V_t indicate void and total volume, respectively. Background absorbance signals before V_0 are not shown fully. G/T/S/K/U refers to the subunits GPAA1/PIGT/PIGS/PIGK/PIGU. The position of each subunit was separately determined by comparing the complex with singly expressed subunits. An asterisk indicates a minor contaminant. Molecular weights of the protein markers are indicated on the right. Uncropped images are included in **Supplementary Data 2**.

2. One question that remains to be answered is whether or not the conditions used for purification would preserve the activity of human GPI-T. While developing a true in vitro GPIT activity assay is likely beyond the scope of this manuscript, it would be helpful for the authors comment briefly on why they feel the purified reconstituted complex is active and their structural data captured an active state of GPI-T.

We thank the reviewer for raising this point. The functionality of purified GPI-T complex in detergents has been a long-standing question and indeed a very important aspect of GPI-T studies. We hope to develop GPI-T assays in future studies. For now, we have added the following texts in the Discussion to address this comment.

“...For the same reason, it remains to be investigated if our method of GPI-T purification preserves its enzymatic activity. However, the fact that the complex showed a Gaussian peak on gel filtration, the enzyme’s ability to trap an endogenous GPI ligand at its active site after lengthy solubilization/detergent exchange/chromatography steps, and the consistency between the structural observation and previous biochemical knowledge of the complex suggest functional relevance of our structure.”

3. The number of mutants that the authors have generated for their cell-based activity assay is very impressive. One caveat of the approach used is the authors cannot completely rule out the loss or reduction in activity is due to low expression of the mutant subunit and/or inability of the mutant subunit to be integrated into the GPI-T complex.

We thank the reviewer for this insightful comment. We have added the following paragraph in Discussion to address this point.

“The interpretation of the apparent activity from the GPI-AP reporter assay is worth discussing. Assays for wild-type and mutants should ideally be performed under conditions where activity responds linearly with the enzyme loading for normalization reasons. In cell-based assays where concentrations of enzyme and substrates are less controllable, the actual activity could be either over-estimated or under-estimated. Overestimation could happen if the expression of a mutant is higher than the wild-type. However, this limitation is less problematic for a complex than for single-component enzymes. Unlike single-component enzymes, overexpression of any individual subunits at the endogenous background level of other subunits would not increase the functional concentration of the complex (unless the subunit increases the level of other subunits by transcriptional/translational regulation or stability reasons). Thus, the change of apparent activity should be attributed to the mutation of interest, except for mutations that compromise complex assembly in which case the defects could be under-evaluated owing to the possible concentration-dependent promotion of complex formation. On the other hand, underestimation could happen if the expression of a mutant is lower than the wild-type. While it is true that not all mutants were expressed at an equilevel and hence this scenario remains possible, we note that the staining of the GPI-AP reporter was counted only for the TGP-positive cells (as an indication of the expression of the subunit of interest) (**Supplementary Data 1**). Although some TGP-positive signals may be due to free-TGP from degradation, a cleaved C-terminal TGP does not necessarily translate to a degraded and nonfunctional subunit. Further, we found that the levels of degradation were minor, and were comparable between the wild-type and mutants for particular subunits according to the in-gel fluorescence results (**Supplementary Data 2**). Finally, we should note that the reasons for loss-of-function mutants for the PIGK and GPI-binding sites were interpreted as being critical for substrate binding, but the possibility exists that they may compromise activity by affecting inter-subunit interactions, especially for those at the shared cavity.”